



# An expert survey on chamber measurement techniques for methane fluxes

Katharina Jentzsch[1,2], Lona van Delden[1], Matthias Fuchs[3], Claire C. Treat[1]

[1]Alfred Wegener Institute (AWI) Helmholtz Centre for Polar and Marine Research, Potsdam, Germany
[2]Institute of Environmental Science and Geography, University of Potsdam, Potsdam, Germany
[3]Renewable and Sustainable Energy Institute, University of Colorado Boulder, Boulder, CO, USA

*Correspondence to*: Katharina Jentzsch (katharina.jentzsch@awi.de)

**Abstract.**

Methane is an important greenhouse gas but the magnitude of global emissions in particular from natural sources remain highly uncertain. To estimate methane emissions on large spatial scales, methane flux data sets from field measurements collected and processed by many different researchers must be combined. We hypothesize that considerable uncertainty might be introduced into such data synthesis products by the many different approaches used to collect, process and quality control chamber measurements of methane fluxes within the flux community. Existing guidelines on chamber measurements promote

more standardized measurement and processing techniques but to our knowledge, so far, no study has investigated which methods are actually used within the flux community. Therefore, we aimed to identify major differences between the approaches for chamber methane fluxes used by different researchers.

We conducted an expert survey to collect information on chamber-based methane flux measurements, including field sites, research questions, measurement setups and routines as well as data processing and quality control of data. We received

responses from researchers in North America, Europe, and Asia which indicated that 80% of respondents have adopted high-frequency, multi-gas analyzers with most measurement times falling between 2 and 5 minutes. Most but not all of the respondents use recommended chamber designs, including such as airtight sealing, fans, and a pressure vent. We asked about the participants' approach to quality control and presented a standardized set of methane concentrations from observed flux measurements, then included this information for flux calculations. The responses showed broad disagreement among the

experts on processes resulting in nonlinear methane concentration increases. Based on the expert responses, we estimated an uncertainty of 28% introduced by different researchers deciding differently on discarding vs. accepting a measurement when processing a representative data set of chamber measurement. Different researchers choosing different time periods within the same measurement for flux calculation caused an additional uncertainty of 17%. Our study highlights the need to understand drivers of the patterns visible from high-resolution analyzers and standardized procedures and guidelines for future chamber

methane flux measurements. This is highly important to reliably quantify methane fluxes all over the world.



## 1 Introduction

Methane ($CH_4$) is an important greenhouse gas with 45 times the global warming potential of carbon dioxide ($CO_2$) on a 100-year timescale (Neubauer, 2021). However, emission estimates differ largely between "top-down" atmospheric measurement inversions and "bottom-up" approaches using data-constrained or process-based models (Kirschke et al., 2013; Saunois et al.,
2020). Natural emissions, especially bottom-up wetland emissions, are the largest source of uncertainty to the global $CH_4$ budget due to the poorly constrained areal extent of wetlands and other methane-producing ecosystems like lakes, streams, and reservoirs, highly uncertain $CH_4$ process parameterization, and a lack of validation data sets (Melton et al., 2013; Saunois et al., 2020).

Despite more than thirty years of chamber-based methane flux measurements from wetland ecosystems (Bartlett &
Harriss, 1993; Harriss et al., 1985), developing large-scale methane validation data sets remain challenging. One approach has been to create synthesis datasets of measurements made by multiple researchers using chamber-based methane flux measurements (Kuhn et al., 2021; Olefeldt et al., 2013; Treat et al., 2018). These data sets should capture the high spatial and temporal variability in natural $CH_4$ emissions with small-scale spatial changes in environmental and ecological conditions (Frenzel and Karofeld, 2000; Laine et al., 2007; Moore and Knowles, 1990; Waddington and Roulet, 1996). This is commonly
achieved with the closed-chamber technique in which a chamber is placed on top of the soil and the change in gas concentrations in the chamber headspace is monitored over time to estimate the exchange of $CH_4$ between soil and atmosphere on the microscale (e.g. Livingston and Hutchinson, 1995). The rate of change in gas concentrations after correcting for the Ideal Gas Law is then used to compute the flux of $CH_4$ through the surface area covered by the chamber (Holland et al., 1999). Two approaches for measuring methane gas concentrations in the chamber are typically used: manual sampling and in-line
gas analyzers. Manual sampling of gas concentrations involves extracting gas samples from the chamber headspace in regular time intervals using syringes and subsequently analysing them for $CH_4$ concentrations on a gas chromatograph. A linear fit is then usually applied to the $CH_4$ concentration measurements over time and its slope is used as the flux estimate after correction for the pressure and temperature inside the chamber (Holland et al., 1999). Manual sampling of the chamber headspace is typically characterized by a low sampling frequency which requires a relatively long chamber closure time. The considerations
here are balancing the time need to get a detectable change in CH4 concentrations versus shorter measurement times to reduce chamber effects (Holland et al., 1999).

With the advances in laser spectroscopy, manual sampling is increasingly replaced by continuously circling chamber air through an in-line gas analyser which performs high-frequency (>1 Hz), high-accuracy, real-time measurements of the $CH_4$ concentration. Through their portability and with reduced measurement times, such multigas analysers have opened new
possibilities, particularly for the analysis of key trace gases like $CH_4$ and $N_2O$. At the same time, the high frequency and high accuracy of the concentration measurements uncover chamber artefacts and events of ebullitive $CH_4$ emission that are superimposed on the signal of natural diffusion of $CH_4$ between soil and atmosphere. Leakage of gas from the chamber (Hutchinson and Livingston, 2001), a saturation effect changing the concentration gradient between soil and chamber





headspace over time (Livingston and Hutchinson, 1995), and natural $CH_4$ ebullition (Strack et al., 2005) as well as ebullition
triggered by the chamber placement can all lead to a deviation of the concentration change from the linear increase expected
for a constant diffusive flux. These observations call for a reassessment of the measurement, processing, and quality control
(QC) approaches to minimize the influence of chamber effects on the flux estimate.

Besides the general lack of validation data sets, existing data sets that combine flux data collected by different
researchers likely include additional uncertainty due to the variety of measurement and processing approaches used. Several
studies have assessed the difference in flux estimates resulting from different chamber setups (Pihlatie et al., 2013; Pumpanen
et al., 2004) and from different data processing approaches such as using nonlinear as compared to linear fits to the gas
concentration measurements over time (Forbrich et al., 2010; Healy et al., 1996; Pirk et al., 2016). Such experimental and
modelling studies have contributed to several guidelines for chamber measurements that were published as an attempt to
establish a more standardized protocol for flux measurements (e.g. Clough et al., 2020; de Klein and Harvey, 2012; Fiedler et
al., 2022; Maier et al., 2022).

While guidelines outlining best measurement practices for chamber measurements provide a well-founded summary
of methods recommended to collect high-quality flux data, they cannot control the approaches that are actually used by the
individual researchers. Discussions with other researchers who use chambers to measure methane fluxes revealed that a wide
range of measurement and processing approaches are still being used by different researchers. At the same time, method
descriptions for chamber-based data collection are often lacking detailed reports on the chamber design, flux calculation and
QC methods, which makes a comprehensive comparison of chamber-based data sets highly uncertain.

Given that the recommendations outlined in guidelines for chamber measurements have significant effects on the
magnitude of CH4 fluxes measured, we need to know how widely implemented these recommendations are and where key
differences and knowledge gaps remain. Gathering scientific and technical insights from experts is necessary to move beyond
established theoretical knowledge and can offer further evidence to aid in decision-making (Morgan, 2014). Several studies
have recently used expert assessments to gain valuable insights into current climate change related topics (Macreadie et al.,
2019; Rosentreter et al., 2024; Schuur et al., 2013). In this study, we use expert judgement derived from a questionnaire to
identify the methods for chamber measurements, processing, and QC of CH4 fluxes that are actually currently used within the
flux community and to assess resulting uncertainties.

This study aims to identify the main discrepancies between the measurement approaches used by different researchers
and their potential effect on the resulting flux data sets. Our objectives were to (1) provide an overview of the chamber designs,
measurement setups and routines, flux calculation and QC approaches that are currently used within the community to estimate
CH4 fluxes as extracted from an expert survey; (2) estimate the variance in QC approaches between different researchers when
processing a representative data set of chamber measurements, (3) identify major sources of uncertainty resulting from the
variety of measurement, calculation, and QC approaches used within the flux community. Our study raises awareness for and
provides a starting point to reduce uncertainties and differences in chamber methods used within the flux community – a





potentially considerable but often neglected source of error in synthesis studies that combine flux data sets collected and processed by different researchers.

## 2 Methods

For this study, we conducted and evaluated an expert survey that consisted of two parts – the first part asking questions about the professional background of the participants and the field sites as well as the measurement, calculation and QC approaches that they use for their own chamber measurements of $CH_4$ fluxes and the second part being an exercise on visual QC of a given set of chamber measurements. Experts were required to have a minimum of one field season of measurements. Experts were solicited through professional networks using emails and conference poster presentations, including the Permafrost Carbon

Network (pcn.org), C-Peat network, ICOS, and through identification of experts not represented to increase the number and geographic background of the participants. Altogether, 46 experts were contacted by email. The survey was estimated to take 40 minutes to complete and the survey language was English. The survey was administered using LimeSurvey (Community Edition Version 5.6.68+240625). Survey participants were asked if they wished to be acknowledged or remain anonymous. The complete, archived survey can be found in Supplement 1 and the survey responses are provided in Jentzsch et al. (2024b).

**2.1 Survey part 1 – The survey participants and their chamber measurements**

In the first, informative part of the survey, we gathered information on the measurement, data processing and QC approaches that the participants use for their own chamber measurements. This part of the survey contained 40 questions of different formats, consisting of 20 multiple choice questions of which 7 were yes/no questions and all with the option to elaborate the response in a short free text comment, five questions asking for multiple short text, 14 free text questions and one image file

upload. To assess the professional background of the group of participants we asked about their professional status, the country of their home institute as well as their educational and scientific background. Multiple responses were allowed. For an overview of the area of application of chamber $CH_4$ flux measurements, we included questions on the participants' research questions and the regions and ecosystem types they usually work in. Questions on the chamber dimensions, the chamber equipment, measurement instruments, as well as photos thereof, together with questions on the measurement procedure and additional

variables monitored showed us the variety of measurement setups used. Additionally, we asked the participants to describe their approaches for flux calculation, quality control, and uncertainty estimation of the flux estimates.

## 2.2. Survey part 2 - Visual quality control of a standardized data set

To more directly compare the different interpretation of chamber data that leads to the discrepancies in measurement setups, data processing, and QC techniques as identified in the first survey part, we provided a standardized set of chamber

measurements for QC by the survey participants and extrapolated the responses to a larger, representative data set. This second part of the survey included both qualitative and quantitative responses.



The standardized set of chamber $CH_4$ fluxes was based on 12 selected chamber measurements from Siikaneva bog (61°50'N, 24°12'E), Southern Finland, in summer 2021 and summer and fall 2022. The measurements were done with a manual chamber with a volume of 36 l, equipped with a cooling system to keep the chamber temperature close to constant, two fans to mix the air inside the chamber, and a small opening for pressure equilibration. For the measurements, the chamber was placed on collars that were permanently installed in the ground. In 2021, the connection between chamber and collar was sealed with a rubber skirt and in 2022 the rim between chamber and collar was filled with water to make the connection air tight. The gas concentrations in the chamber were measured with an in-line gas analyzer at a frequency of 1 Hz. Besides measurements showing a linear increase in $CH_4$ concentration over time, we included examples showing a variety of deviations from the linear increase expected for constant diffusive wetland $CH_4$ emissions.

The survey questions for visual QC included the concentrations of $CH_4$ over time as well as the simultaneously measured concentrations of $CO_2$ and $H_2O$ in the chamber, a photo of the chamber, and a description of the measurement setup as well as information on dominant vegetation and water table depth at the measurement plot, date and time of the measurement, transparent vs. opaque chamber, gas analyzer model and a photo of the measurement plot (Figure A1a). We asked the participants if they would keep the respective measurement for flux calculation or if they would discard it and why they would do so (Figure A1b). If they decided to keep the measurement, we asked them to select the part of the measurement that they would use to calculate the $CH_4$ flux and submit this for a calculation of $CH_4$ flux based on their response as the quantitative portion of the response.

## 2.3. Statistical analyses

### 2.3.1 Cleaning of the data set

We anonymized the survey by separating the demographic information including the country of the home institute, the scientific background, the highest education level, the time since PhD completion, and the current professional role of the participants from each other and from the rest of the survey responses. We furthermore removed the question for specific research sites before publishing the data and replaced two names of specific research sites given as a description of the main study regions by terms for a larger region. In one response, we removed the name of another researcher mentioned by one of the participants.

We harmonized and/or categorized some free text responses including the responses on the chamber shape, the chamber area, chamber volume, the closure time of the chamber, and the frequency of the gas concentration measurements inside the chamber. We used chamber volume and chamber area to calculate the effective chamber height. We corrected some obvious writing mistakes throughout the survey as part of the standardization. In the visual QC part of the survey, we standardized the information on the exclusion of the beginning of the measurement from flux calculation as well as the length of the excluded time period. We also adjusted the responses to questions whether to keep or to discard a measurement in the





visual QC part when the free text responses clearly revealed that the wrong box was ticked by mistake. We set the $CH_4$ flux to zero in two cases where survey participants clearly stated in their free text responses that this is what they would do.

**2.3.2 Evaluating the visual QC exercise**

We quantitatively and qualitatively evaluated the responses to the visual QC portion of the survey. We summarized the free text responses in the visual QC part for the qualitative responses related to the reasons for keeping or discarding a flux. Then, we numerically evaluated the visual QC performed on the 12 example measurements. This allowed us to quantify the variation in fluxes due to quality control and differences in fitting approaches among researchers. For this, we calculated $CH_4$ fluxes for

each researcher for each of the 12 selected QC measurements using the time periods selected by the researcher.

We used a standard linear fitting approach for the flux calculation, accounting for differences in temperature and pressure among the measurements (Holland et al., 1999). The Ideal Gas Law was used to convert the rate of change in $CH_4$ concentrations ($\frac{dc_{CH_4}}{dt}$) in ppm s$^{-1}$ to the molecular $CH_4$ flux ($F_{CH_4}$) in mol m$^{-2}$ s$^{-1}$ for each measurement example $i$ (i=1,…,n, where n=12) and each survey participant $j$ (j=1,…,m, where m=36).

$$F_{CH_4\,i,j} = \frac{dc_{CH_4}}{dt}\bigg|_{i,j} \times 10^{-6} \times \frac{p}{R \times T_i} \times \frac{V_i}{A},$$

where $p$ represents the standard atmospheric pressure of 101325 Pa, $T$ (degrees K) is the mean of the temperature inside the chamber during the closure, $A$ is the surface area of the chamber in m$^2$. $V_i$ is the volume of the chamber used in measurement $i$, calculated by $V_i = A \times h_i$, where $h_i$ is the effective height of the chamber headspace during measurement $i$ (in m), calculated as the mean of the height above the soil surface or vegetation cover that was measured at three points around the chamber for

each measurement plot. $R$ is the Ideal Gas Constant of 8.314 kg m$^2$ mol$^{-1}$ K s$^{-2}$. We then converted the molecular $CH_4$ flux to the more commonly used mass flux of $CH_4$ using the molar mass of $CH_4$ of 16.04 g mol$^{-1}$. For each measurement example and each participant, $\frac{dc_{CH_4}}{dt}$ was estimated as the slope of a linear fit (lm function from stats package in R version 4.3.0) to the $CH_4$ concentrations within the time period selected by the researcher. For reasons of consistency, we used a linear fit even in the 12 cases that a participant suggested to use a nonlinear fit instead (7% of the total of 173 times that start and end times for flux

calculation were given by the participants). When a measurement was accepted by an expert but no start and end time was given for flux calculation, we estimated the flux based on the entire chamber measurement.

We used the fluxes calculated from the quantitative responses to assess the uncertainty due to different researchers processing the measurement data that is the uncertainty due to different researchers (1) selecting time periods for flux calculations, and 2) deciding whether to keep or to discard a measurement (quality control). In a representative data set of 788

chamber measurements, collected at Siikaneva bog in 2021 and 2022 (Jentzsch et al., 2024a), we visually identified and categorized the following eight classes of measurements based on the shape of the $CH_4$ concentrations measured in the chamber headspace over time: "Linear increase", "Linear decrease", "Nonlinear increase – decreasing slope", "Nonlinear increase –

increasing slope", "Initial jump", "Jump(s)", "Inconsistent trend", and "No trend" (Table 1). From the Siikaneva data set, we selected 12 measurement examples so that each measurement class was represented at least once in the visual QC exercise.

For each measurement class, we estimated the uncertainty introduced by different researchers choosing different time periods within the same measurement for flux calculation using the coefficient of variance (CV) across the fluxes calculated for each survey participant. To extrapolate this uncertainty to a representative data set, we calculated the weighted sum of the CVs based on the relative occurrence of each measurement class within the Siikaneva data set. To assess the uncertainty due to quality control we extrapolated the percentage of measurements kept for flux calculation to a representative data set for each

participant, again using the relative occurrence of each measurement class within the Siikaneva data set. We then calculated the CV between the percentages of measurements kept across all survey participants.

## 3 Results

A total number of 36 expert researchers participated in the survey. All of them completed the survey parts on demographic information and their field sites for flux measurements. Most (35 participants) answered the questions concerning their flux

measurement setup, and 30 responded about their flux calculation and QC approach. Participation decreased to 28 experts for the visual quality control exercise and an additional two dropped out after the second example measurement, resulting in a survey completion rate of 72%.

### 3.1 Results of: Survey part 1 - The survey participants and their chamber measurements

### 3.1.1 Demography

The participants work for universities (25 participants), research centers (11 participants), or companies (one participant) that are located in North America, central and northern Europe, and eastern Asia (Figure 1a). Most (89%) of the participants have a PhD title, 41% of whom completed their PhD within the last seven years, 25% between 7 and 15 year ago, and 34% more than 15 year ago (Figure 2a). Nearly all (94%) of the participants are researchers, two of whom are PhD students (Figure 2b). One participant each specified their current position as Bachelor student, professor, leader in industry, coordinator, and

consultant, respectively. With 58%, the majority of the survey participants has a background in Geosciences, followed by Biology (25%), Ecology (11%), Meteorology (8%), Environmental sciences (6%), and Physics (6%). One participant each has a background in forestry, biogeosciences, and agricultural sciences. Half of the participants (52%) are part of one or several of the flux networks and databases FluxNet, ICOS, AmeriFlux, OzFlux/TERN, European Fluxes Database Cluster, and LTER.



**Figure 1: Countries of the main institutes (a) and of the research sites (b) of the participants.**

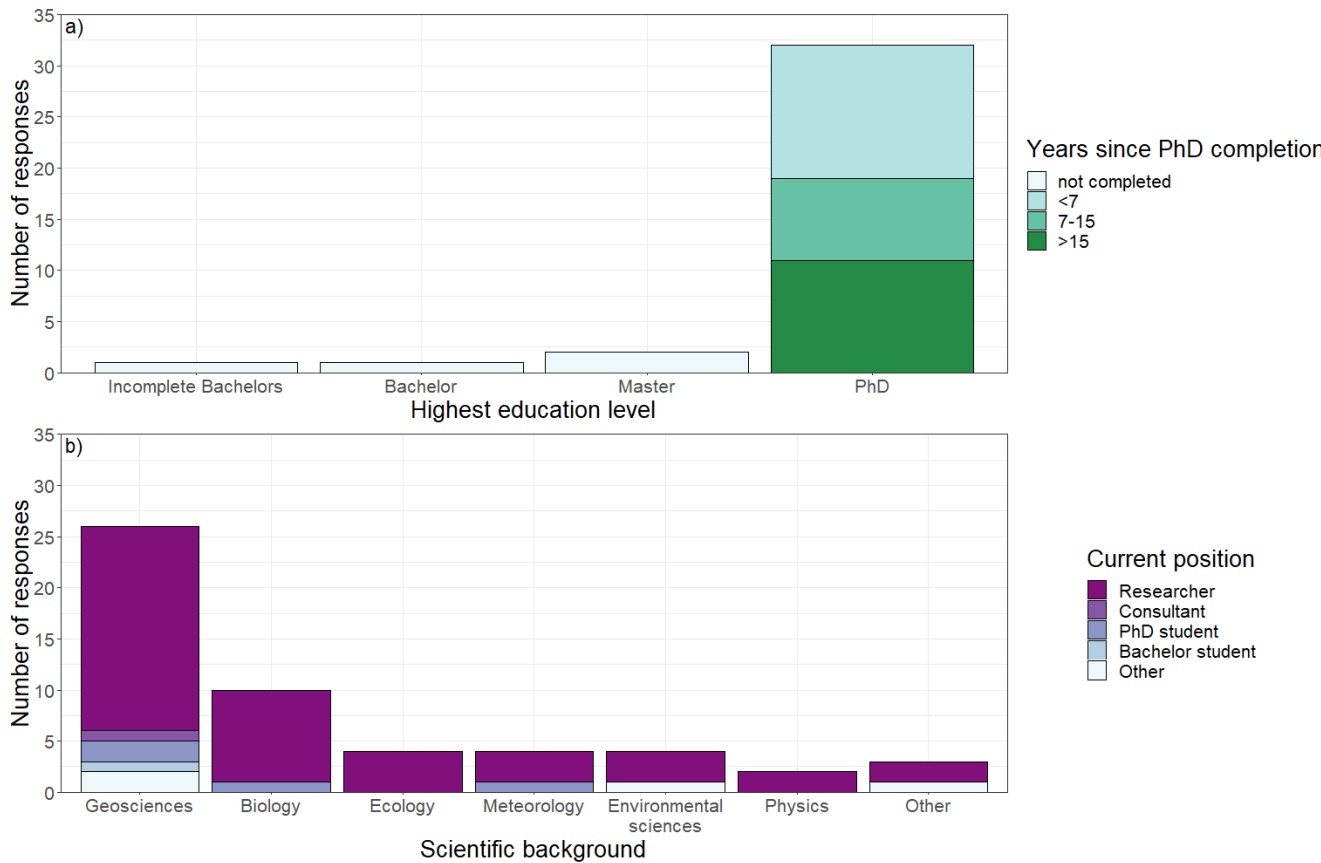

**Figure 2: Histograms of the highest education level of the participants, split by the years since their PhD completion (a) and of their scientific background by current position (b).**

### 3.1.2 Flux measurement sites

Most (83%) of the participants do field measurements in the country of their home institute, among them all participants working for institutions in Asia, Canada, Finland, Norway, Denmark, Austria, and the United Kingdom (Figure 1b). Four participants additionally reported field measurements in Greenland and one participant in Ghana, Costa Rica, and Senegal, which were not among the countries of home institutes of the participants. Six participants from the US, Germany, and Sweden had their main research sites in Canada, Finland or Greenland, according to their research questions and ecosystems of interest. The majority (83%) of the participants focus their research on peatlands and wetlands, mainly fens or bogs (50%), and littoral wetlands (31%) (Figure 3). A few (14%) of the participants measured in (semi-)arid regions, upland areas, and at sites with mineral soil instead of or in addition to wetlands. Some (33%) of the participants explicitly mentioned field measurements in permafrost-affected landscapes; similarly, 33% of the participants explicitly mentioned that they measure in "northern", "boreal", "arctic", or "subarctic" regions and 6% measure in "alpine" or "subalpine" terrain. Some (25%) of the participants





do aquatic measurements and 19% measure at anthropogenically managed sites such as on agricultural land, in drained and in rewetted peatlands.

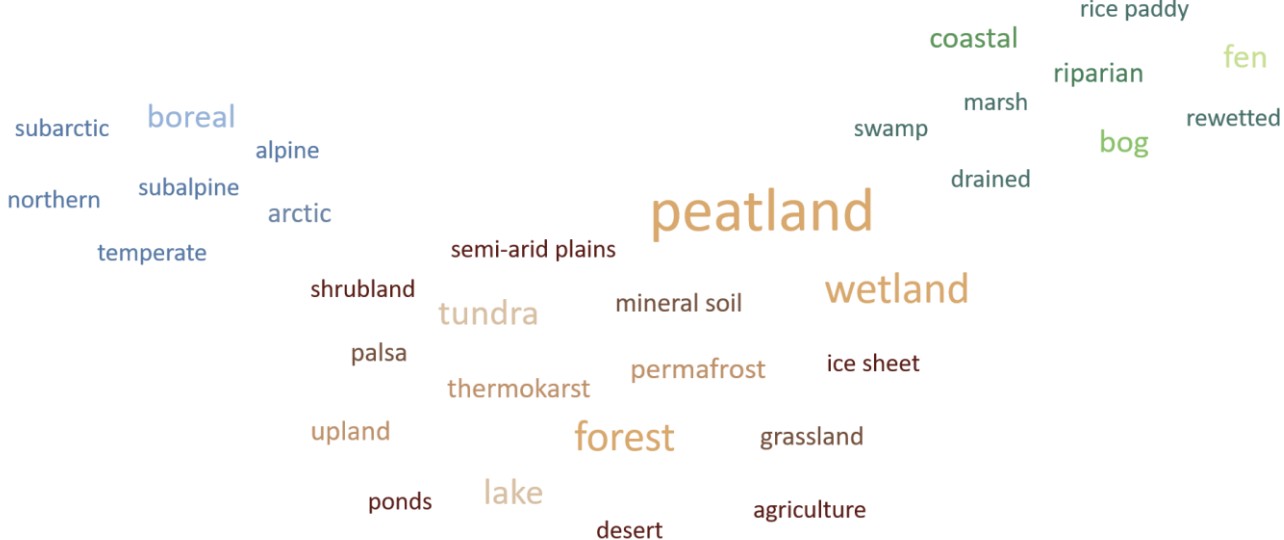

**Figure 3: Word cloud of the study areas with the brownish, middle part of the word cloud representing the studied ecosystem types,**
**the blue part in the top left corner showing the climatic zones of the study sites, and the green part in the top right specifying the types of wetlands and peatlands that are researched by the participants.**

The overarching research goals that the survey participants address with their flux measurements are to better understand the processes involved in greenhouse gas cycling, to better understand and quantify the effect of changes on greenhouse gas
dynamics, to estimate greenhouse gas budgets, and to research the methodology for gas flux measurements. To investigate the environmental and ecological controls on the greenhouse gas exchange is the main goal of 28% of the participants, mainly in peatlands and wetlands and considering environmental conditions, vegetation properties, and the microbial community among others. Specific ecosystems researched by two participants are rice paddies and reed ecosystems. The main aim of 53% of the participants is to understand and/or to quantify the effect of natural and anthropogenically induced change on greenhouse gas
dynamics. The changes considered involve climate change, more specifically, warming, vegetation changes, elevated atmospheric $CO_2$ concentrations, permafrost thaw, and intensifying disturbances, such as wildfires, as well as peatland management, land-use change, and oil and gas exploration. Estimating greenhouse gas budgets is the goal of 22% of the participants but goals differed in spatial and temporal scales from annual budgets of northern ecosystems to budgets of wetlands, microseepage, i.e. diffusive $CH_4$ fluxes over productive hydrocarbon basins, as an estimate of natural geologic $CH_4$
emissions, or permafrost and periglacial ecosystems, including thermokarst lakes, thawing permafrost peatlands, and degrading subaqueous permafrost. One participant uses the flux measurements to research methodologies for gas flux measurements,





investigating their accuracy, minimum detectable fluxes, curve fitting approaches, as well as engineering challenges around automation and minimizing measurement artefacts.

### 3.1.3 Flux measurement setup

The participants use different instrumental setups to measure gas fluxes with the chamber technique (for examples see figure A2). Of the 33 participants who responded to the question, 82% use manual chambers and 33% use chambers that open and close automatically. The shape of the chamber differs between participants, with 64% using square or rectangular chambers and 55% using cylindrical chambers. The volume of the chambers ranges between 8 and 1800 L with a median of 64 L and an interquartile range (IQR) of 105 L. 93% of the chambers used are smaller than 260 l (Figure A3). The participants equipped

their chambers with different devices and use varying approaches to sample the chamber air and measure the concentrations of different greenhouse gases (Figure 4). The majority of the participants (80%) use an in-line gas analyser for on-site, continuous, and high-frequency measurements of the gas concentrations. All but one of them use a closed sample loop which returns the air to the chamber after circulation through the gas analyser. Fewer (26%) of the participants manually sample the chamber air for analysis of a gas chromatograph. One participant uses open-path LI-COR gas analysers installed inside a large

chamber. The gas analysers used by the survey participants continuously measure the gas concentrations at high frequencies between five times per second and once every 15 s and therefore require shorter closure times of 0.5 to 12.5 min compared to the closure times of 16 to 50 min used when manually sampling the chamber air every 4 to 10 min (Figure A4). Most survey participants (58%) calibrate their gas analysers once per year and 24% do so once before each measurement campaign. A few (10%) of the participants calibrate the gas analyser less often e.g. when serviced every 1 to 3 years and 12% calibrate more

frequently ranging from weekly to daily to calibration after each flux measurement. As required for the survey participation, all participants measure the $CH_4$ concentrations in the chamber, 97% additionally measure $CO_2$ and 33% measure the $N_2O$ concentrations. One participant each additionally measures BVOC and Ethane.

Most participants (83%) record the temperature inside the chamber and 29% measure the pressure inside the chamber with measurement frequencies ranging from every second to once per chamber closure. Half of the participants use a vent to

keep the air pressure inside the chamber close to ambient conditions and thereby prevent pressure changes from altering the natural gas flux. Different methods for pressure equilibration used were a hole in the chamber that is sealed after chamber placement, explicitly mentioned by two participants, and a long line of tubing that is constantly open to the atmosphere allowing for pressure equilibration while preventing that too much ambient air enters the chamber, explicitly mentioned by one participant. The majority of the participants (80%) uses a fan to mix the air inside their chamber. Three participants argued

that the air flow from circulation through a closed loop with the gas analyser was sufficient to mix the chamber air, particularly in small chambers. Some (20%) of the participants use a cooling system to prevent the temperature of the air inside the chamber from rising too much above the ambient air temperature. Types of cooling systems mentioned were Peltier elements, circulation of the chamber air through a tank filled with ice-water, fans circulating the cold air from ice packs placed inside the chamber, and an opaque or reflecting cover on the chamber. Most participants (66%) place their chamber on top of a collar that they





inserted into the soil between one hour and one year before the measurement. The majority of the participants (63%) aim to make the connection between the chamber and the collar or the soil gastight by using one or several types of sealing. Besides gaskets and water seals, a plastic sheet weighed down by a chain, a stocking filled with sand, and foam in the collar groove were mentioned as sealing methods.

      The participants take various measures to minimize potential disturbances to their chamber measurements. For wet,

terrestrial sites, 28 participants stand on more stable ground while measuring, either by using permanently or temporarily installed boardwalks or wooden boards or by choosing a drier patch or a rock to stand on. Six participants furthermore mentioned that they make sure not to walk close to the measurement plots by using automated chambers or walking rules supported by warning tape. For aquatic measurements, participants avoid anthropogenic disturbance of the sediment and thus of the gas release by pulling the chamber into its measurement location with a rope or sitting in a boat while measuring. In

addition, careful placement of the chamber, training of those who measure, maintenance of collars and sealing, and carefully keeping the vegetation away from the chamber sides were used to minimize disturbances to the chamber measurements.

      Additional variables are recorded by many participants alongside the chamber measurements in order to identify potential disturbances or to explain the observed gas fluxes (Figure 5). To characterize the soil and hydrological conditions, variables like C and N content, pH value, reduction-oxidation potentials, soil moisture, soil or water temperature and water

table depth are measured by 97% of the participants. Meteorological conditions are documented by 88% of the participants recording variables like atmospheric temperature, atmospheric pressure, relative humidity, vapor pressure deficit, rainfall, wind direction, wind speed and photosynthetically active radiation, in particular when partitioning the measured $CO_2$ fluxes into gross primary productivity and ecosystem respiration. The vegetation cover of the measurement plots is characterized by 15% of the participants through assessing the plant species composition, measuring plant height, specific leaf area, or leaf dry

matter content or estimating the leaf area index of the individual plant species or the moss cover by species. In cold regions or seasons, thaw depth, active layer depth, snow depth and/or frost depth are recorded by 9% of the participants.


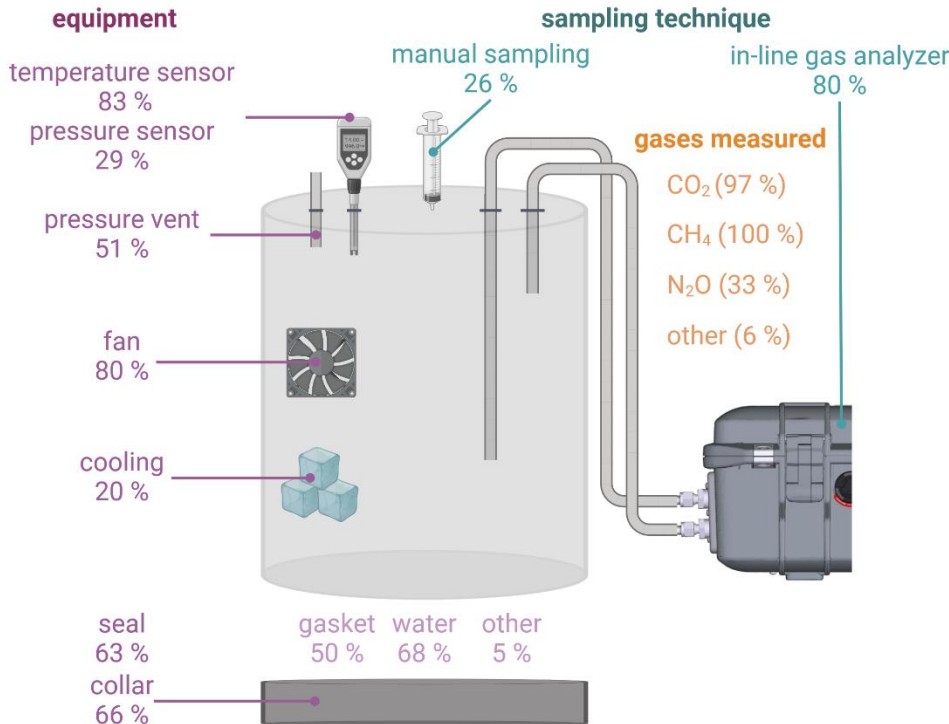

**Figure 4: Schematic chamber set up including the percentage of survey participants using certain types of chamber equipment, sampling approaches for the gas concentrations and measuring different greenhouse gases.**





**Figure 5: Ancillary data recorded alongside the gas fluxes.**

### 3.1.4 Flux calculation and QC approaches

The qualitative responses on calculation approaches for $CH_4$ fluxes revealed that slight differences in the measurement routines and processing techniques might result in considerable differences between the $CH_4$ fluxes derived by different researchers from the same chamber measurement. Gas fluxes are generally estimated from chamber measurements as the slope of the change in gas concentration over the time of the chamber closure and accounting for the water vapor concentrations, the temperature and the pressure inside the chamber as well as for the chamber dimensions. This approach was modified by the survey participants mainly through selecting a time period of each chamber measurement for flux calculation, choosing a fit function to estimate the change in concentration over time, and determining the accuracy of the temperature and pressure



correction by selecting a measurement frequency for the two variables or deciding to use standard values instead. To avoid initial disturbances caused by the chamber placement, a large share of the participants (43%) excludes the beginning of each chamber measurement from their flux calculation ranging from the first 5 s to the first 170 s of the measurement with a median of discarding the first 30 s of each measurement. The majority of the participants (70%) use a linear relationship of the change in gas concentration over time from each chamber measurement. Some (23%) of the participants use both a linear fit as well as the initial slope of an exponential fit, either deciding for one based on the goodness of the fit or using the range of the two slopes as uncertainty estimate. One participant each uses an exponential fit on all chamber measurements, considers the total change in gas concentration as the difference between the gas concentrations at the start and at the end of the chamber closure, or averages multiple linear fits on a one-minute window moving over the measurement at steps of 10 s.

Of the 17% of participants who do not measure the temperature inside their chamber, one participant explicitly mentioned that they assume standard temperature in their flux calculations, while two participants use air temperature measurements. The two thirds of survey participants who do not measure the air pressure inside their chamber similarly assume standard pressure or a constant value adapted to their measurement site or use ambient pressure measurements instead. At least 43% of the participants use one temperature value, either measured at the beginning or at the end of the chamber closure, or averaged over the whole chamber measurement for each flux calculation. On the contrary, two participants explicitly stated that they correct each gas concentration measurement individually for the chamber temperature and/or pressure measured at or interpolated to the same frequency as the concentration measurements. The majority of the participants (90%) use self-written scripts and functions for their flux calculation while 20% of the participants at least partly use existing and published R or Matlab scripts.

Various approaches for QC of the time series of gas concentrations recorded during the chamber measurements were mentioned by the participants. Most participants (60%) manually check each of their chamber measurements, while 17% of the participants use an automated QC procedure and 23% used a combination of both manual and automated diagnosis. A majority (53%) of the participants use the goodness of fit to evaluate the quality of their flux estimates, with the $R^2$ and RMSE (root mean square error) being the preferred quality measures used by 75% and 38% of the participants, respectively. Fixed cut-off values given by 7 participants for the $R^2$ ranged between 0.75 and 0.95 and a RMSE threshold value of 0.5 was given by one participant. Two participants furthermore only accept measurements with a significant change in the $CH_4$ over time with p values smaller than 0.05 and 0.001, respectively. In addition, 17% of the participants visually inspect the time series of $CH_4$ concentrations resulting from each chamber measurement, 7% check that the starting concentration in the chamber does not deviate too much from ambient conditions, and 13% each consider instrument diagnostics for their quality assessment or identify cases of low or no $CH_4$ flux for special treatment. Most participants (43%) discard between 2 and 5% of their data based on their QC procedure. For 17% of the participants, their QC protocol leads to an exclusion of up to 1% of their measurements while two participants discard up to 100% of their data (Figure A5).

One third of the participants assesses the uncertainty of each individual flux estimate, most of them (80%) based on the uncertainty of the fit parameters expressed as $R^2$, RMSE, or standard deviation of the slope and the two remaining





participants based on the difference between the slopes derived from a linear compared to an exponential fit or the variation in several one-minute linear fits in a moving time window.

### 3.2 Results of: Survey part 2 - Visual quality control of a standardized data set

#### 3.2.1 General trends

The visual QC exercise revealed how the measurement examples were handled by the survey participants, their explanations of nonlinear behaviour, as well as how much the flux estimates were affected by the time period chosen for flux calculation depending on the behaviour of the $CH_4$ concentrations (Table 1). The majority of the participants (91%) decided to keep the measurements that showed a linear increase in $CH_4$ concentrations for flux calculation. Due to the linear behaviour these flux estimates were least affected by the time period that was chosen for the linear fit, showing in a low variation of 2.5% between
the flux estimates (Figure 6). Although the flux estimates from measurements that showed a consistent but nonlinear trend in $CH_4$ concentrations were strongly affected by the time period chosen for flux calculation, most participants (79%) also kept these measurements. More than half of the participants kept the linear increase in $CH_4$ concentrations following an initial jump in $CH_4$ concentrations resulting in little variation between the flux estimates similar to the measurements with the linear increase persisting over the entire closure time. Measurements showing a linear decrease, low variation, or one or several
intermittent jumps in $CH_4$ concentrations, were kept by less than 50% of the participants although the variation in the resulting flux estimates between the survey participants was in a similar range or even lower than for the nonlinear measurement examples. Less than one quarter of the participants kept the measurement example showing a changing trend in $CH_4$ concentrations over the course of the measurement. The resulting flux estimates varied strongest between the participants with a coefficient of variance which exceeded the mean flux value including both positive and one negative flux estimate. Overall,
participants kept between 33-100% of the VQC flux examples for varying reasons although one participant discarded all fluxes.

**Table 1: VQC exercise (blue shading): Fluxes in mgCH$_4$ m$^{-2}$ d$^{-1}$ calculated for each chamber measurement based on the time periods chosen for flux calculation by each survey participant. Coefficients of variance (CV) are given for each example measurement and measurement class across the participants. In the table, "NR" means that a participant did not respond to the respective**
**measurement example so neither kept not discarded the measurement. "D" indicates that a participant discarded a measurement example. Participants who gave no response to more than one measurement example are marked with "n.p." for "not participated". Percentage of measurements kept for flux calculation (purple shading): Percentage of measurement examples within the visual QC exercise kept for flux calculation by participant (Kept$_{VQC}$, relative to total number of measurement examples that a participant responded to), measurement example (Kept$_{ex}$), and measurement class (Kept$_{class}$). Extrapolation to Siikaneva data set and**
**uncertainty estimates (red shading): Extrapolation of the visual QC results to the entire Siikaneva data set through weighting by the frequency of occurrence of each measurement class in the data set. Weighting factors are derived as the relative occurrence of the respective measurement class in the Siikaneva data set (788 measurement in total) divided by the number of occurrences in the visual QC exercise.**





| Measurement class | Linear increase | | Nonlinear increase – decreasing slope | | | Initial jump | Jump(s) | | Nonlinear increase – increasing slope | Inconsistent trend | Linear decrease | No trend |
|---|---|---|---|---|---|---|---|---|---|---|---|---|
| Weight [%] | 468 (59.4) | | 144 (18.3) | | | 66 (8.4) | 62 (7.9) | | 25 (3.2) | 16 (2.0) | 4 (0.5) | 3 (0.4) |
| Mes ID | VQC1 | VQC2 | VQC4 | VQC5 | VQC9 | VQC7 | VQC8 | VQC12 | VQC10 | VQC11 | VQC3 | VQC6 |
| Weight [%] | 29.7 | 29.7 | 6.1 | 6.1 | 6.1 | 8.4 | 4.0 | 4.0 | 3.2 | 2.0 | 0.5 | 0.4 |

| Mes ID / Participant | VQC1 | VQC2 | VQC4 | VQC5 | VQC9 | VQC7 | VQC8 | VQC12 | VQC10 | VQC11 | VQC3 | VQC6 | Kept$_{VQC}$ [%] | Kept$_{sii}$ [%] |
|---|---|---|---|---|---|---|---|---|---|---|---|---|---|---|
| 1 | 67.31 | 3204.96 | 68.56 | 171.47 | 1102.58 | 351.70 | 2183.91 | 87.58 | 137.75 | 127.25 | -8.99 | 14.17 | 100 | 100 |
| 2 | 66.80 | 3169.76 | D | D | 1049.99 | 377.13 | D | D | 131.51 | D | D | D | 42 | 77 |
| 3 | 68.68 | 3367.11 | 73.95 | 169.61 | 1036.42 | 371.20 | D | 88.95 | 158.75 | -18.64 | -10.78 | 13.96 | 92 | 96 |
| 4 | 67.02 | 3294.94 | 66.92 | 171.47 | 1790.25 | 350.85 | 3117.04 | 482.53 | 127.40 | D | -11.16 | D | 83 | 98 |
| 5 | 67.02$^a$ | 3166.90$^a$ | 32.80$^a$ | 67.90$^a$ | 1751.88$^a$ | D | D | D | 116.73$^a$ | D | -11.16$^a$ | 0 | 67 | 81 |
| 6 | 66.34 | 3156.77 | 65.24$^{n.l.}$ | 61.26 | 1535.05 | D | D | 275.62 | D | D | D | D | 50 | 52 |
| 7 | 67.17 | 3067.51 | 61.86 | D | 1057.30 | D | 2031.93 | D | 133.38 | D | D | D | 50 | 79 |
| 8 | 67.02 | 3169.76 | D | D | 1091.37 | 378.93 | D | D | 116.73 | D | -11.16 | D | 50 | 78 |
| 9 | 67.02 | 3169.76$^{n.l.}$ | 71.13 | 67.90$^{a, n.l.}$ | 1751.88$^a$ | D | D | D | D | D | -11.08 | 0 | 58 | 79 |
| 10 | 67.02 | 3304.26 | 75.34 | D | D | 380.34 | D | D | 158.75 | D | D | D | 42 | 77 |
| 11 (n.p.) | NR | NR | NR | NR | NR | NR | NR | NR | NR | NR | NR | NR | NA | NA |
| 12 (n.p.) | NR | NR | NR | NR | NR | NR | NR | NR | NR | NR | NR | NR | NA | NA |
| 13 (n.p.) | NR | NR | NR | NR | NR | NR | NR | NR | NR | NR | NR | NR | NA | NA |
| 14 | D | 3166.90$^a$ | 77.53 | 180.71 | 4216.12 | D | D | D | 168.07 | D | D | D | 42 | 51 |
| 15 (n.p.) | NR | NR | NR | NR | NR | NR | NR | NR | NR | NR | NR | NR | NA | NA |
| 16 | 67.02$^a$ | 3166.90$^a$ | 72.72 | 110.49 | 3553.67 | 380.34 | 2016.94 | 482.53$^a$ | 125.25 | D | -11.16$^a$ | D | 83 | 98 |
| 17 (n.p.) | D | NR | NR | NR | NR | NR | NR | NR | NR | NR | NR | NR | NA | NA |
| 18 | 66.14 | 3293.94 | 32.80$^a$ | 67.90$^a$ | 1122.55 | 357.19 | D | 482.53$^a$ | 131.32 | D / 9.58$^a$ | D | D | 71 | 94 |
| 19 | 67.02 | 3169.76 | 34.38 | 69.94 | 1790.25 | 416.49 | 3117.04 | 482.53 | 116.73 | 9.58 | -11.16 | 9.95 | 100 | 100 |
| 20 (n.p.) | NR | NR | NR | NR | NR | NR | NR | NR | NR | NR | NR | NR | NA | NA |
| 21 | 67.31 | 3229.48 | 66.81 | 171.47 | 1084.14 | 357.19 | D | 88.36 | 125.19 | 127.25 | -8.92 | 14.17 | 92 | 96 |
| 22 | 71.78 | 3301.49 | 79.90$^{n.l.}$ | 67.90$^a$ | 4973.26 | D | D | D | 75.82$^{n.l.}$ | D | D | D | 50 | 81 |
| 23 | 66.34 | 3141.12 | 70.84 | 130.21$^{n.l.}$ | 1680.02 | 371.20 | 2016.94 | 88.95 | 142.01 | -6.17 | -14.85 | 14.16 | 100 | 100 |
| 24 | 67.31 | 3399.49 | 73.94 | 169.61 | 5122.50 | 380.27 | D | 273.84 | 74.68 | D | -10.76 | D | 75 | 94 |
| 25 | D | D | D | D | D | D | D | D | NR | D | D | D | 0 | 0 |
| 26 | 67.02 | 3204.96 | 76.38 | D | D | D | D | D | D | D | -11.16 | D | 33 | 66 |
| 27 (n.p.) | NR | NR | NR | NR | NR | NR | NR | NR | NR | NR | NR | NR | NA | NA |
| 28 | 67.02 | 3286.36 | 53.58$^{n.l.}$ | 119.60 | 2482.08 | D | D | 275.62 | D | D | D | D | 50 | 82 |
| 29 | 67.17 | 3178.85$^{n.l.}$ | D | 66.39$^{n.l.}$ | 1764.55 | 377.13 | 1723.41 | D | 127.24 | 8.45 | D | 12.77 | 75 | 90 |
| 30 | D | 3358.45 | D | D | 1088.47 | 346.32 | D | D | D | D | D | 8.51 | 33 | 45 |
| 31 | 67.02$^a$ | 3312.29 | 72.72 | 180.71 | 5750.52 | 180.71$^a$ | D | 482.53$^a$ | 116.73$^a$ | 138.74 | D | D | 75 | 95 |
| 32 (n.p.) | 67.02$^a$ | 3304.26 | NR | NR | NR | NR | NR | NR | NR | NR | NR | NR | 100 | NA |
| 33 (n.p.) | NR | NR | NR | NR | NR | NR | NR | NR | NR | NR | NR | NR | NA | NA |
| 34 | 69.67 | 3304.26 | 65.05 | 138.58 | 1065.04 | 380.34 | D | 275.62 | D | D | -15.16 | 9.95 | 75 | 91 |
| 35 (n.p.) | NR | NR | NR | NR | NR | NR | NR | NR | NR | NR | NR | NR | NA | NA |
| 36 | 67.31 | 3321.94 | 73.94 | D | 1124.44 | D | D | 88.95 | 120.74$^{n.l.}$ | D | D | D | 50 | 79 |
| Mean±SD | 67.36±1.18 | 3238.93±84.93 | 65.07±14.52 | 121.29±48.77 | 2129.75±1490.01 | 374.37±19.88 | 2315.31±564.43 | 282.58±172.84 | 126.57±23.50 | 49.50±68.31 | -11.36±1.89 | 9.76±.55 | Weighted sum: | 80±23 |




| CV [%] | 2 | 3 | 22 | 40 | 70 | 5 | 24 | 61 | 19 | 138 | 17 | 57 | 17 | 28 |
|---|---|---|---|---|---|---|---|---|---|---|---|---|---|---|
| CV [%] | 2.5 | | 44 | | | 5 | 42.5 | | 19 | 138 | 17 | 57 | | |
| Kept_ex [%] | 86 | 96 | 81 | 69 | 88 | 62 | 27 | 54 | 76 | 29 | 50 | 38 | | |
| Kept_class [%] | 91 | | 79 | | | 62 | 40 | | 76 | 29 | 50 | 38 | | |

[a] accepted but no time period was given for flux calculation. The flux was therefore estimated based on the entire measurement.

[n.l.] Flux was estimated based on a linear fit although the participant suggested to use a nonlinear model instead.

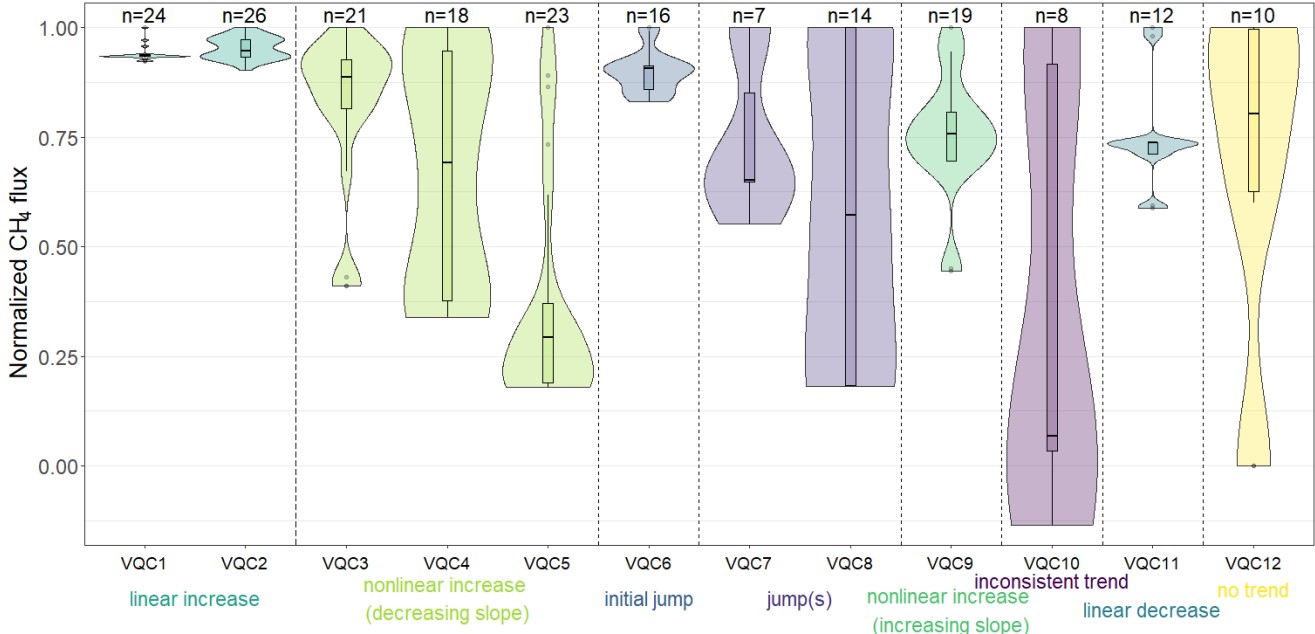

**Figure 6: Distribution of flux estimates for each measurement example in the visual QC exercise (VQC1 – VQC12) by measurement class. Flux estimates are normalized within each measurement example. The violins are scaled to all have the same maximum width. The number of survey participants (n) who contributed a flux estimate to the respective measurement example by selecting a time period for flux calculation is given on top of each violin.**

### 3.2.2 Linear increase

Two example measurements included in the visual QC exercise showed a linear increase in $CH_4$ concentrations over the entire time of the chamber closure. In the first example (measurement ID VQC1, Figure 7), the $CH_4$ concentrations increased by only 0.4 ppm from a starting concentration of 2.1 ppm to a concentration of 2.5 ppm at the end of the measurement while in the second example (measurement ID VQC2, Figure 8) the starting concentration of 3.8 ppm was higher and increased by 38.2 ppm to reach 42.0 ppm over the course of the measurement. The $CO_2$ concentration in the chamber decreased during both measurements by 13.4 ppm and 9.1 ppm, respectively. The starting concentration of $H_2O$ in VQC2 was more than 10 times higher than for VQC1 and decreased strongly over the course of the measurement with an abrupt decrease at around 50 s after chamber closure, while $H_2O$ concentrations increased slightly in VQC1.





The participants described the trend in $CH_4$ concentrations in both VQC1 and VQC2 as a linear increase which they explained by net $CH_4$ production and diffusive emission. The $CH_4$ flux in VQC2 was additionally classified as large with one participant concluding that the measurement plot was a "hotspot" for $CH_4$ emission. For both VQC1 and VQC2, some participants additionally noticed slight deviations from the linear behaviour of the $CH_4$ concentrations. Minor jumps in the $CH_4$ concentration in VQC1 were mentioned by 17 participants (61%), which they related to $CH_4$ ebullition (9 participants),

insufficient mixing due a defective fan (3 participants), wind (1 participant), wind-induced pressure changes (1 participant), changes in atmospheric pressure influencing the ground diffusion rates and/or atmospheric pressure gradient (1 participant), boundary layer disturbance (1 participant), leakage (2 participants), disturbance (1 participant) caused by chamber placement or footsteps (1 participant). For VQC2, half of the participants pointed out a decrease in the slope of $CH_4$ concentrations starting between 250 and 260 s after the chamber closure, 21% of whom also noticed a simultaneous decrease in the slope of

$CO_2$ concentrations. As explanations the participants mentioned saturation of the chamber headspace decreasing the concentration gradient over time (5 participants), a build-up of pressure (2 participants) potentially due to a defective pressure valve towards the end of the measurement (1 participant), a change in temperature over the course of the measurement (1 participant), or a small leak (1 participant) probably combined with windy conditions (1 participant). Many participants furthermore discussed the change in $CO_2$ and $H_2O$ concentrations over the time of the chamber closure. For VQC1, three

participants mentioned that the $CO_2$ and $H_2O$ concentrations show a linear change, two of whom concluded that there was no air leaking from the chamber. Three participants on the other hand were concerned about the $H_2O$ measurements due to the high and increasing concentrations, and due to an assumed saturation and therefore decreasing slope towards the end of the measurement. Leakage from the chamber was suspected by three participants, two of whom explained this presumption with vegetation overgrowing the collar and one with the use of a less airtight rubber seal as opposed to a water seal. For VQC2,

18% of the participants picked up on the drop in $H_2O$ concentrations occurring around 40 s after the chamber closure, 40% of whom additionally mentioned a simultaneous change in the slope of $CO_2$ concentrations. Their reasoning included water condensing on the chamber walls and changing light conditions. Few participants decided to discard the two measurement examples. Measurement VQC1 was discarded by 4 participants (14%) suspecting $CH_4$ ebullition or stating that the starting concentrations of $CO_2$ were too high above ambient concentrations or that all chamber measurements generally need to be

shaded. One participant excluded VQC2 due to an assumed saturation effect and one additional participant mentioned ebullition and a high initial concentration of $CH_4$ as potential reasons to exclude the measurement from flux calculations. 86% and 89% of the participants decided to keep VQC1 and VQC2 for flux calculation, respectively, due to the consistent linear increase in $CH_4$ concentrations without clear indications of significant disturbances or any malfunctioning of the instruments. For VQC1, the participants further supported their decision with the linear change in $CO_2$ and $H_2O$ concentrations making

leakage from the chamber unlikely as well as with near-ambient $CH_4$ concentrations at the measurement start. For both VQC1 and VQC2 most participants who gave start and end times for flux calculation chose the middle part of the measurement, discarding the beginning and the end without mentioning a specific reason. The remaining participants considered the $CO_2$ and/or $H_2O$ concentrations in their choice of the time period for curve fitting. For VQC1, three participants chose the beginning



of the measurement only, resulting in slightly higher flux estimates, two of whom assumed that $H_2O$ saturation diminished the

increase in $CH_4$ concentrations towards the end of the measurement. For VQC2, some participants acknowledged the strong

drop in $H_2O$ concentrations. Having no further information on potential reasons three of them decided not to let this unexpected

behaviour in $H_2O$ concentrations make them discard the $CH_4$ measurements while other participants reacted by excluding the

time of the drop in $H_2O$ concentrations from their calculation of the $CH_4$ flux through either using the part of the measurement

after the drop (7 participants) or before the drop (1 participant). 61% of the 23 participants who entered start and end times for

flux calculation discarded the end of the measurement where $CH_4$ and $CO_2$ concentrations increased at a lower rate, resulting

in slightly higher flux estimates above 3200 $mgCH_4\,m^{-2}\,d^{-1}$. Two participants suggested to use a nonlinear fit which one of

them specified as exponential.

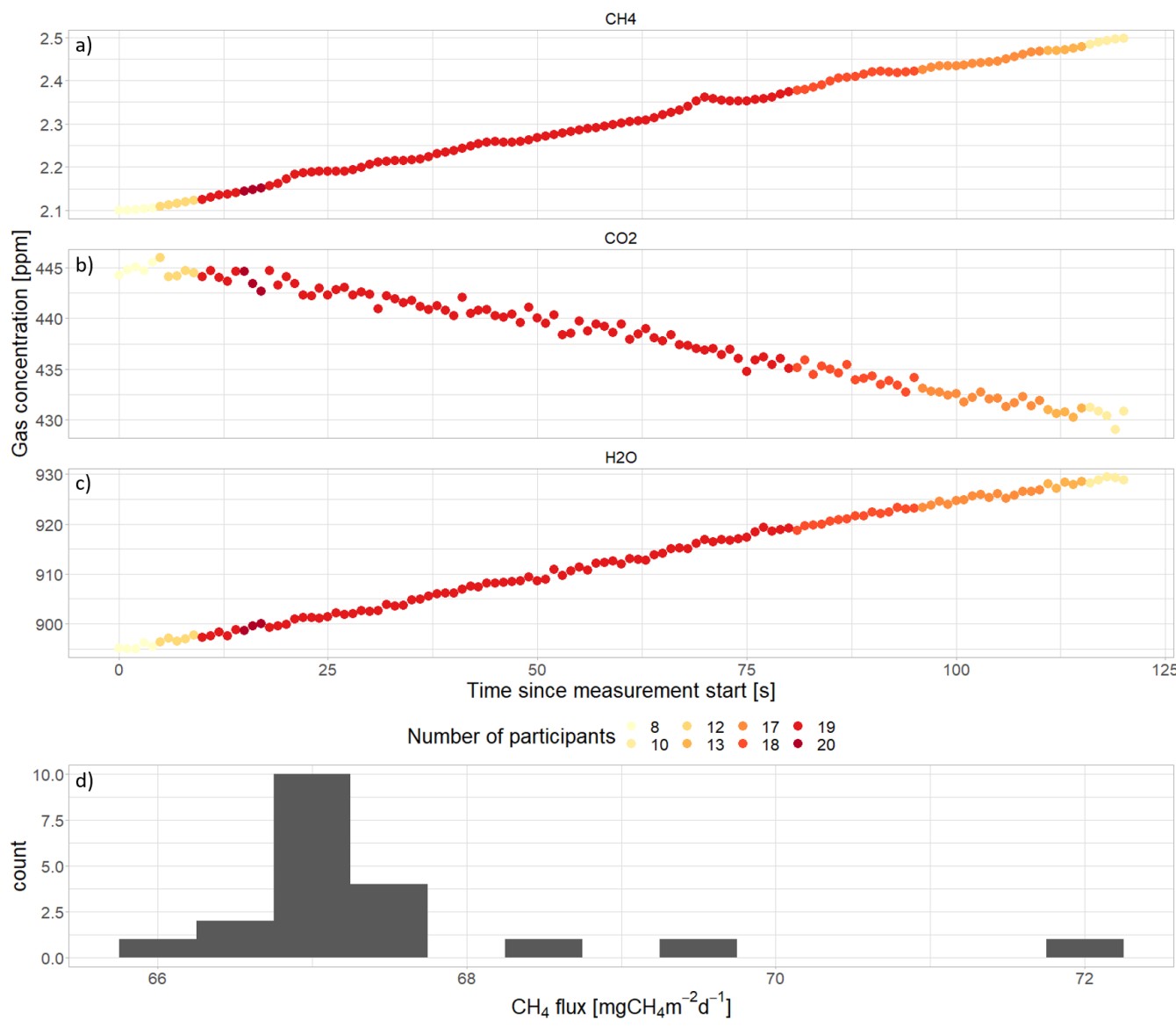


**Figure 7: Measurement example VQC1 of a small linear increase in CH₄ concentrations over the time of the chamber closure. Simultaneous measurements of CH₄ (a), CO₂ (b), and H₂O (c) concentrations over time during chamber closure with the colours of the data points indicating how many participants included the respective data point in the time period that they chose for flux calculation. Only the responses by participants who gave start and end times for flux calculation were considered for this figure.**

**Histogram of CH₄ fluxes calculated based on the time period chosen for flux calculation by the participants (d).**





**Figure 8: Measurement example VQC2 of a strong linear increase in CH₄ concentrations over the time of the chamber closure.**
**Simultaneous measurements of CH₄ (a), CO₂ (b), and H₂O (c) concentrations over time during chamber closure with the colours of**
**the data points indicating how many participants included the respective data point in the time period that they chose for flux**
**calculation. Only the responses by participants who gave start and end times for flux calculation were considered for this figure.**
**Histogram of CH₄ fluxes calculated based on the time period chosen for flux calculation by the participants (d).**

### 3.2.3 Nonlinear increase – decreasing slope

In the visual QC exercise, we included three examples of measurements that feature a nonlinear increase in CH₄ concentrations
during the chamber closure with the rate of increase flattening out over time. Two examples show a small nonlinear increase





in $CH_4$ concentrations (measurement IDs VQC4 and VQC5, Figures 9 and 10) simultaneous with linearly decreasing $CO_2$ concentrations. $H_2O$ concentrations increased over the time of the chamber closure in VQC4 but decreased in VQC5. The third example (measurement ID VQC9, Figure 11) shows a stronger increase in $CH_4$ concentrations with intermittent jumps, linearly increasing $CO_2$ concentrations and $H_2O$ concentrations that fluctuate without a clear trend.

The participants classified the $CH_4$ measurements in VQC5 as a small flux that resulted from a balance between $CH_4$ production and oxidation while VQC9 was identified as large emission indicating a $CH_4$ hotspot. The majority of the participants (85%, 85%, and 81%) discussed the nonlinear behaviour of the $CH_4$ concentrations in VQC4, VQC5, and VQC9, respectively, offering various explanations for the decreasing rate of increase over time that were mainly related to chamber saturation, chamber leakage or an initial disturbance (Table 2).

Most participants (9, 10, and 8) suspected a saturation of the chamber headspace, while two participants stated that saturation was unlikely to be reached during a measurement as short as VQC4 (330 seconds) and one participant explicitly mentioned that the changing slope in VQC9 did not look like a saturation effect. Just as many participants (9) suggested leakage through a weakening seal as the reason for the decreasing slope in VQC4, as supported by the simultaneously decreasing slope in $CO_2$ and $H_2O$ concentrations while other participants explicitly stated that $CO_2$ and $H_2O$ concentrations did not indicate a

leak in this measurement. Due to the consistently linear $CO_2$ concentrations in VQC5 and VQC9, only one participant each suspected leakage during these measurement examples. For VQC4, three participants further suspected that the high $H_2O$ concentrations at the end of the measurement influenced the $CH_4$ measurements, for example through condensation inside the chamber or in the gas flow line, one participant suggested a varying performance of the chamber fan, and two participants assumed that the nonlinearity was a phenomenon specific to *Sphagnum* moss. One participant more generally suggested that

vegetation effects changed over the course of the measurement due to changing light conditions, affecting the $CO_2$ and $H_2O$ concentrations in VQC4 and the $CH_4$ concentrations in VQC5.

        Besides a saturation effect or a weakening seal that would cause a decreasing slope in $CH_4$ concentrations towards the end of the measurement, many participants (3, 6 and 8) suggested that an initial disturbance such as ebullition triggered by the chamber placement had caused the stronger increase in the beginning of measurement examples VQC4, VQC5, and VQC9.

For VQC9, 31% of the participants additionally pointed out minor fluctuations superimposed on the overall nonlinear increase in $CH_4$ concentrations. Two thirds of them referred to the fluctuations as minor ebullition events while the others suggested episodic leakage from the chamber potentially caused by gusts of wind lifting the chamber sides or a malfunctioning pressure gauge. One participant pointed out that the $CH_4$ fluctuations cooccurred with fluctuations in the $H_2O$ concentrations and therefore suspected an instrument issue that could be related to spikes in the instrument cavity pressure.

The nonlinearity in the $CH_4$ concentrations resulted in 15%, 31%, and 12% of the participants deciding to discard the entire measurement example VQC4, VQC5, and VQC9, respectively (Table 2). The reasons mentioned for the exclusion of the measurements again reflected the different interpretations of the participants on which part of the measurement represented the real flux. This disagreement shows less strongly in the range of flux estimates since participants who suspected an initial disturbance of the measurement disproportionately often discarded the entire measurement as they assumed that an initial




disturbance would also affect the remaining part of the measurement. For VQC4 and VQC5, all 54% of the participants who provided start and end times for flux calculations agreed that the beginning of the measurement should be used for or at least be included in the flux calculation, with three participants suggesting a nonlinear fit for both measurement examples. This resulted in smaller ranges of flux estimates compared to VQC9 (Table 2) which instead reflects the fundamentally different interpretations among the participants on which part of the measurement should be used for flux calculation. Here, half of the

21% of participants who gave start and end times for flux calculation chose a later part of the measurement where $CH_4$ concentrations appeared linear over a longer time period. This resulted in lower flux estimates (between 1000 and 1200 mg$CH_4$ m$^{-2}$ d$^{-1}$) compared to the flux estimates larger than 3500 mg$CH_4$ m$^{-2}$ d$^{-1}$ derived for the one quarter of participants who instead chose the beginning of the measurement (Figure 11).

**Table 2: Explanations for the nonlinear increase in $CH_4$ concentrations, reasons to discard, and reasons and ways to keep measurements showing an increase in $CH_4$ concentrations as given by the participants. The responses were categorized based on the free text entries for measurement examples VQC4, VQC5, and VQC9. The number of responses given in the respective category are provided in brackets.**

| Explanations for nonlinearity | Reasons to discard | Reasons to keep |
|---|---|---|
| Saturation (23) | Saturation (2) | A nonlinear fit can be used (9) |
| | Nonlinearity – no steady state reached (3) | A (linear) part of the curve can still be used (41) |
| Initial disturbance (16) | Initial disturbance biases flux later on (2) | |
| Bad seal / Leakage from the chamber (8) | Bad seal / Leakage from the chamber (4) | |
| Unsure (12) | Unclear which part of the measurement represents the real flux (3) | No clear disturbance of the measurement (9) |
| Changing environmental conditions (1) | Changing environmental conditions (1) | Linear trend in $CO_2$ concentrations (5) |
**Figure 9: Measurement example VQC4 of a small nonlinear increase in CH₄ concentrations with decreasing slope over the time of the chamber closure. Simultaneous measurements of CH₄ (a), CO₂ (b), and H₂O (c) concentrations over time during chamber closure with the colours of the data points indicating how many participants included the respective data point in the time period that they chose for flux calculation. Only the responses by participants who gave start and end times for flux calculation were considered for this figure. Histogram of CH₄ fluxes calculated based on the time period chosen for flux calculation by the participants (d).**





**Figure 10: Measurement example VQC5 of a small nonlinear increase in CH₄ concentrations with decreasing slope over the time of the chamber closure. Simultaneous measurements of CH₄ (a), CO₂ (b), and H₂O (c) concentrations over time during chamber closure**
**with the colours of the data points indicating how many participants included the respective data point in the time period that they chose for flux calculation. Only the responses by participants who gave start and end times for flux calculation were considered for this figure. Histogram of CH₄ fluxes calculated based on the time period chosen for flux calculation by the participants (d).**





**Figure 11: Measurement example VQC9 of a strong nonlinear and jumpy increase in CH₄ concentrations with decreasing slope over the time of the chamber closure. Simultaneous measurements of CH₄ (a), CO₂ (b), and H₂O (c) concentrations over time during chamber closure with the colours of the data points indicating how many participants included the respective data point in the time period that they chose for flux calculation. Only the responses by participants who gave start and end times for flux calculation were considered for this figure. Histogram of CH₄ fluxes calculated based on the time period chosen for flux calculation by the participants (d).**



### 3.2.4 Jump(s) at the beginning or in the middle of measurements

In our visual QC exercise, we included three example measurements that showed a relatively linear increase in $CH_4$ concentrations that was interrupted by one (VQC8, Figure 12 and VQC7, Figure 13) or more (VQC12, Figure 14) sudden increases in the concentration. In one, this occurred at the beginning (VQC7) and in the others, the middle of the measurement

(VQC8, VQC12). In examples VQC8 and VQC7, these sudden jumps appeared in all three gases with $CH_4$ and $H_2O$ concentrations showing a sudden increase while $CO_2$ concentrations dropped simultaneously. In VQC12, on the contrary, $CO_2$ and $H_2O$ showed no equivalent to the jumps in the $CH_4$ concentration. In VQC8, a strong decrease in $CH_4$ concentrations directly followed the sudden increase, while in VQC7 and VQC12 the concentrations continued to increase at a lower rate starting close to the high concentration level after the jump.

Nearly all (100%, 65%, and 92%) of the participants mentioned the jump(s) in $CH_4$ concentration when discussing the measurement examples VQC8, VQC7, and VQC12, respectively. For all measurement examples, the majority of these participants explained their observation with episodic events of ebullitive $CH_4$ emission (VQC8: 65%, VQC7: 88%, VQC12: 92%) with only one participant each suggesting a malfunctioning of the gas analyser as a reason for the sudden increase in $CH_4$ concentrations in VQC7 and VQC12. Some (35%) of the participants assuming ebullition stated that the ebullition event

in VQC8 was caused by a disturbance and all agreed that the chamber placement caused the ebullition for VQC7. Only one participant (5%) mentioned anthropogenic disturbance as the reason for the ebullition events in VQC12. For VQC8, 12% of the participants pointed out the sudden changes in $CO_2$ and $H_2O$ concentrations along with the jumps in $CH_4$. Reasons mentioned by one participant each were a malfunctioning of the gas analyser and an overpressure caused by the bubble release while another participant suggested the release of gas bubbles with high $CH_4$ but low $CO_2$ concentrations as a natural cause

for this observation. Similarly, one of the two participants who mentioned the absence of a simultaneous change in the other gases in VQC12, assumed a release of bubbles with high $CH_4$ concentration but $CO_2$ concentrations close to ambient conditions due to the different production depths of the two gases. For VQC8, 41% of the participants discussed the decrease in $CH_4$ concentrations following the assumed ebullition event and suggested leakage of air from the chamber, potentially combined with wind as a potential cause. In the discussion of VQC7, two participants disagreed on the effect of the water table on $CH_4$

ebullition, one mentioning that in the measurement $CH_4$ ebullition was more likely to happen because of the high water table while the other stated that ebullition happened despite the high water table, indicating a fundamentally different understanding of the causes of $CH_4$ ebullition among the participants. Two participants of VQC7 furthermore classified the measurement as an example of strong $CH_4$ emission which they explained by strong anaerobic $CH_4$ production related to the high water table and by the vegetation providing substrate for acetoclastic $CH_4$ production, respectively.

Of the three measurements with jumps in $CH_4$ concentrations that we included in the visual QC exercise, VQC8 raised the most concern with the highest number of participants excluding the example (Table 2) and with the largest variety of reasons mentioned for the discard, including the inconsistent trend in $CH_4$ and $CO_2$ concentrations making them wonder which part of the measurement to use for flux calculation, ebullition affecting the pressure inside the chamber, too much variation in





$CH_4$ and $CO_2$ concentrations even after the jump, chamber leakage and too high initial $CH_4$ and $CO_2$ concentrations. Leakage
was also suggested by one participant for VQC7, who suspected that Sphagnum moss might have obstructed the chamber seal
        with the collar. VQC12 was classified as too short of a measurement by one participant and discarded by another for too high
        initial $CO_2$ concentrations.

        There was disagreement among the participants on whether the remaining part of a measurement after a jump in the
        $CH_4$ concentration could still be used for flux calculation. For 10 of 11 participants discarding VQC7, the main concern was
high concentrations having a lasting effect on the concentration gradient and thus on the diffusive $CH_4$ flux during the rest of
        the measurement while only one of 21 and 15 participants discarded measurements VQC8 and VQC12 for that reason. For
        VQC12, four of the 11 participants who kept the measurement, all of whom also gave start and end times, avoided this problem
        by using the beginning of the measurement before the first jump for flux calculation. On the contrary, for VQC8 and VQC7
        five and 14 of the seven and 15 participants who kept the measurement and/or gave start and end times for flux calculation
decided that the measurement after the jump in $CH_4$ concentrations could still be used for flux calculation, respectively, and
        five participants in VQC12 preferred to use the part between the first two jumps because it showed a longer linear increase.

        The choice of different time periods for flux calculation resulted in two and three different classes of flux magnitudes
        for VQC8 and for VQC7 and VQC12, respectively. The highest flux estimates of more than 3000 $mgCH_4\,m^{-2}\,d^{-1}$,
        483 $mgCH_4\,m^{-2}\,d^{-1}$, and 416 $mgCH_4\,m^{-2}\,d^{-1}$ stemmed from the two, one, and two participants who used the whole measurement
example VQC7, VQC8, and VQC12, respectively, for flux calculation because these estimates also included ebullitive in
        addition to diffusive $CH_4$ emissions, reflecting the general disagreement on whether $CH_4$ ebullition should directly be included
        in the flux estimates derived from chamber measurements. For VQC7 and VQC12, the flux estimates from the participants
        who excluded the jumps in $CH_4$ concentration from the time period for flux calculation can further be split into two classes.
        For VQC7, nine participants excluded only the very beginning of the measurement, while five participants only used a later
part starting at about 50 s into the measurement when $CO_2$ concentrations decreased at a higher rate, resulting in slightly lower
        $CH_4$ fluxes. For VQC12, when excluding the jump in $CH_4$ concentrations the flux estimates were higher for four participants
        who chose the measurement period before the first jump, reaching up to 275 $mgCH_4\,m^{-2}\,d^{-1}$ compared to the five participants
        who chose the longest linear part of the measurement leading to flux as low as 88 $mgCH_4\,m^{-2}\,d^{-1}$. Due to the very linear
        behaviour of the $CH_4$ concentrations following the initial jump and the higher agreement on the time period used for flux
calculation, the CV of 5% for VQC7 was much lower than for the CVs of 24 and 61% for VQC8 and VQC12, respectively.





**Figure 12: Measurement example VQC8 of an overall increase in CH₄ concentrations over the time of the chamber closure after a strong jump in CH₄ concentrations. Simultaneous measurements of CH₄ (a), CO₂ (b), and H₂O (c) concentrations over time during chamber closure with the colours of the data points indicating how many participants included the respective data point in the time period that they chose for flux calculation. Only the responses by participants who gave start and end times for flux calculation were considered for this figure. Histogram of CH₄ fluxes calculated based on the time period chosen for flux calculation by the participants (d).**








**Figure 13: Measurement example VQC7 of a linear increase in CH₄ concentrations of the chamber closure after an initial jump in CH₄ concentrations. Simultaneous measurements of CH₄ (a), CO₂ (b), and H₂O (c) concentrations over time during chamber closure with the colours of the data points indicating how many participants included the respective data point in the time period that they chose for flux calculation. Only the responses by participants who gave start and end times for flux calculation were considered for this figure. Histogram of CH₄ fluxes calculated based on the time period chosen for flux calculation by the participants (d).**



**Figure 14: Measurement example VQC12 of a linear increase in CH₄ concentrations between repeated jumps in the CH₄ concentration over the time of the chamber closure. Simultaneous measurements of CH₄ (a), CO₂ (b), and H₂O (c) concentrations over time during chamber closure with the colours of the data points indicating how many participants included the respective data point in the time period that they chose for flux calculation. Only the responses by participants who gave start and end times for flux calculation were considered for this figure. Histogram of CH₄ fluxes calculated based on the time period chosen for flux calculation by the participants (d).**



### 3.2.5 Nonlinear increase - increasing slope

One example included in the visual QC exercise showed a nonlinear increase in CH$_4$ concentrations over the chamber closure with the rate of increase becoming stronger over time (measurement ID VQC10, Figure 15). 15% of the participants classified the measurement as a diffusive emission of CH$_4$ without mentioning further details while 65% discussed the increasing slope in CH$_4$ concentrations over time, suggesting various reasons that could have caused the observed shape of the curve. The reasons suggested included an initial period of mixing or adjusting, an increase in chamber temperature over time, a disturbance

of the measurement plot, a disturbance of the concentration gradient in the soil during chamber placement, an influence of the chamber on plant-mediated CH$_4$ transport, an incomplete seal of the chamber, incomplete mixing, and an interference with the simultaneously increasing H$_2$O concentrations. Two participants mentioned that they had not seen such a shape in CH$_4$ concentrations from chamber measurements before. Regarding the magnitude of CH$_4$ emissions, three participants pointed out the strong increase in CH$_4$ concentrations despite the relatively low water table, which they related to plant-mediated CH$_4$

transport. One participant further mentioned that also the emission of CO$_2$ was high, indicating warm peat conditions. Two participants mentioned the higher and decreasing CO$_2$ concentrations in the beginning of the measurement which one of them related to the chamber placement, pushing more gases out of the ground. One participant furthermore mentioned that the chamber seal seemed to be intact.

Six participants decided to discard the measurement, three of whom did so because they could not explain the shape of the curve

and stated that the curvature was so strong that the flux estimate would strongly depend on the time period chosen for flux calculation. The three remaining participants mentioned similarly unexpected shapes of CO$_2$ and H$_2$O concentrations, higher H$_2$O concentrations towards the end of the chamber closure which might have interfered with the CH$_4$ measurements, and high initial CH$_4$ concentrations as reasons to discard the measurement. 19 participants kept the measurement for flux calculation. The flux estimates for the 17 participants who gave start and end times for flux calculation strongly depended on

the time period they chose which in turn depended on their interpretation of the measurement resulting in three distinct classes of flux magnitudes. Two participants decided to use the entire measurement, resulting in intermediate flux estimates of 117 mgCH$_4$ m$^{-2}$ d$^{-1}$. The majority (13) decided to remove the first 20 to 120 s of the measurement to keep only the more linear part of the CH$_4$ concentrations in the end, resulting in the highest flux estimates between 125 and 170 mgCH$_4$ m$^{-2}$ d$^{-1}$. The two remaining participants chose only the linear first 60 or 70 s of the measurement for flux calculation resulting in lower flux

estimates of 75 and 76 mgCH$_4$ m$^{-2}$ d$^{-1}$, respectively, due to the lower rate of increase. Two participants suggested to use a nonlinear fit which one of them specified as exponential.

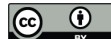



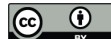

**Figure 15: Measurement example VQC10 of a small nonlinear increase in CH₄ concentrations with increasing slope over the time of the chamber closure. Simultaneous measurements of CH₄ (a), CO₂ (b), and H₂O (c) concentrations over time during chamber closure with the colours of the data points indicating how many participants included the respective data point in the time period that they chose for flux calculation. Only the responses by participants who gave start and end times for flux calculation were considered for this figure. Histogram of CH₄ fluxes calculated based on the time period chosen for flux calculation by the participants (d).**





### 3.2.6 Inconsistent trend

One example included in our visual QC exercise showed an inconsistent trend in $CH_4$ with a change from increasing to decreasing concentrations over the time of the chamber closure (measurement ID VQC11, Figure 16). The survey participants disagreed on the reason for this behaviour of the $CH_4$ concentrations. One part of the participants stated that $CH_4$ oxidation as indicated by the decrease in $CH_4$ concentrations towards the end of the measurement was unexpected and suggested that measurement issues were responsible for the inconsistent trend in $CH_4$ concentrations. They had different opinions however on the timing of the disturbance and therefore on which part of the measurement represented the actual $CH_4$ flux. Some participants suggested an initial disturbance such as $CH_4$ ebullition caused by the chamber placement while others assumed that the measurement was disturbed at a later point by a problem with the $CH_4$ analyzer like saturation of the detector or $H_2O$ interference due to the high concentrations towards the end of the measurement and potentially condensation of water vapor, or leakage or a malfunctioning fan after about 50 s into the measurement.

Most participants (66%) discarded the measurement because they missed a consistent trend of sufficient length in the $CH_4$ concentrations. Since the changing trend was either related to a disturbance or the reason was described as unclear, the participants did not know which part of the measurement to use for the flux calculation. Two participants additionally discarded the measurement because they considered the changes in the $CH_4$ concentration as too close to zero and another participant mentioned that the $CO_2$ and $H_2O$ concentrations did not show a steady trend over time either. Some (23%) of the participants decided to keep the measurement for flux calculation, all of whom provided start and end times for flux calculation as well as one additional participant who was uncertain whether to keep or to discard the measurement. The choice of the time periods used for flux calculation depended on the interpretation of the observed pattern in $CH_4$ concentrations and thus strongly influenced the resulting flux estimate ranging between a $CH_4$ uptake of -19 $mgCH_4\ m^{-2}\ d^{-1}$ to $CH_4$ emissions of up to 139 $mgCH_4\ m^{-2}\ d^{-1}$ and splitting the flux histogram into three distinct modes. Two participants chose to keep the entire measurement, resulting in a small positive flux indicating small net $CH_4$ emission of 8 to 10 $mgCH_4\ m^{-2}\ d^{-1}$. Three participants decided to use the stronger increase in $CH_4$ concentrations in the beginning of the measurement, resulting in the highest $CH_4$ emissions between 127 and 139 $mgCH_4\ m^{-2}\ d^{-1}$ while two participants assumed that $CH_4$ was consumed at the plot, using the later decreasing part of the $CH_4$ concentrations, resulting in negative flux estimates between -6 and -19 $mgCH_4\ m^{-2}\ d^{-1}$. This resulted in the highest CV among the measurement classes, estimated at 138%.





**Figure 16: Measurement example VQC11 of CH₄ concentrations showing an inconsistent trend over the time of the chamber closure. Simultaneous measurements of CH₄ (a), CO₂ (b), and H₂O (c) concentrations over time during chamber closure with the colours of the data points indicating how many participants included the respective data point in the time period that they chose for flux calculation. Only the responses by participants who gave start and end times for flux calculation were considered for this figure. Histogram of CH₄ fluxes calculated based on the time period chosen for flux calculation by the participants (d).**






### 3.2.7 Linear decrease

One of the measurements in the visual QC exercise showed a small linear decrease in $CH_4$ concentrations over time
(measurement ID VQC3, Figure 17). The survey participants largely disagreed on whether this measurement represented a
real $CH_4$ flux. The majority (65%) of the participants assumed real net $CH_4$ uptake due to $CH_4$ oxidation dominating over $CH_4$
production while some (19%) of the participants referred to leakage and too high initial $CH_4$ concentrations in the chamber as
technical problems causing a false apparent uptake of $CH_4$. The remaining 15% of the participants explicitly stated that they
were unsure if the measurement represented a real flux. 23% of the participants more specifically mentioned an inconsistent
trend in the $CH_4$ concentrations referring to three different stages of $CH_4$ flux or nonlinearities at the beginning and at the end
of the measurement. As explanations, they offered initial $CH_4$ ebullition caused by the chamber placement, changes in the
chamber temperature, changes in wind speed combined with chamber leakage, or changes in PAR potentially due to a changing
cloud cover or due to condensation inside the chamber indicated by the trend in $CO_2$ concentrations changing along with the
$CH_4$ trend as well as by high $H_2O$ concentrations.

A slim majority (54%) of the participants discarded the measurement because they did not expect $CH_4$ uptake in the
given environmental (despite the relatively low water table), or because of the inconsistent trend in $CH_4$ concentrations which
makes them unsure which part of the measurement to use for flux calculation, or because of too high initial concentrations of
$CH_4$ and/or $CO_2$, or because they suspected anthropogenic disturbance from footprints and compacted vegetation or leakage.
The flux estimates derived from the start and end times given by 11 of the 12 participants who decided to keep the measurement
(46%) differed between the time periods chosen for flux calculation. While five participants chose the entire measurement,
resulting in intermediate values of $CH_4$ uptake, the remaining six participants chose the time period for curve fitting based on
the $CO_2$ concentrations. The middle part of the measurement with linearly decreasing $CO_2$ concentrations, the beginning of
the measurement with stable $CO_2$ concentrations, and the end of the measurement with linearly increasing $CO_2$ concentrations
were chosen by one, two, and one participant, respectively, while two participants excluded the end of the measurement
resulting in strongly negative, lower negative, stronger negative and intermediate $CH_4$ fluxes, respectively. Overall, the mean
of the flux calculated by the 12 experts keeping this flux was 11.36 $mgCH_4 \, m^{-2} \, d^{-1}$ with a CV of 17%.

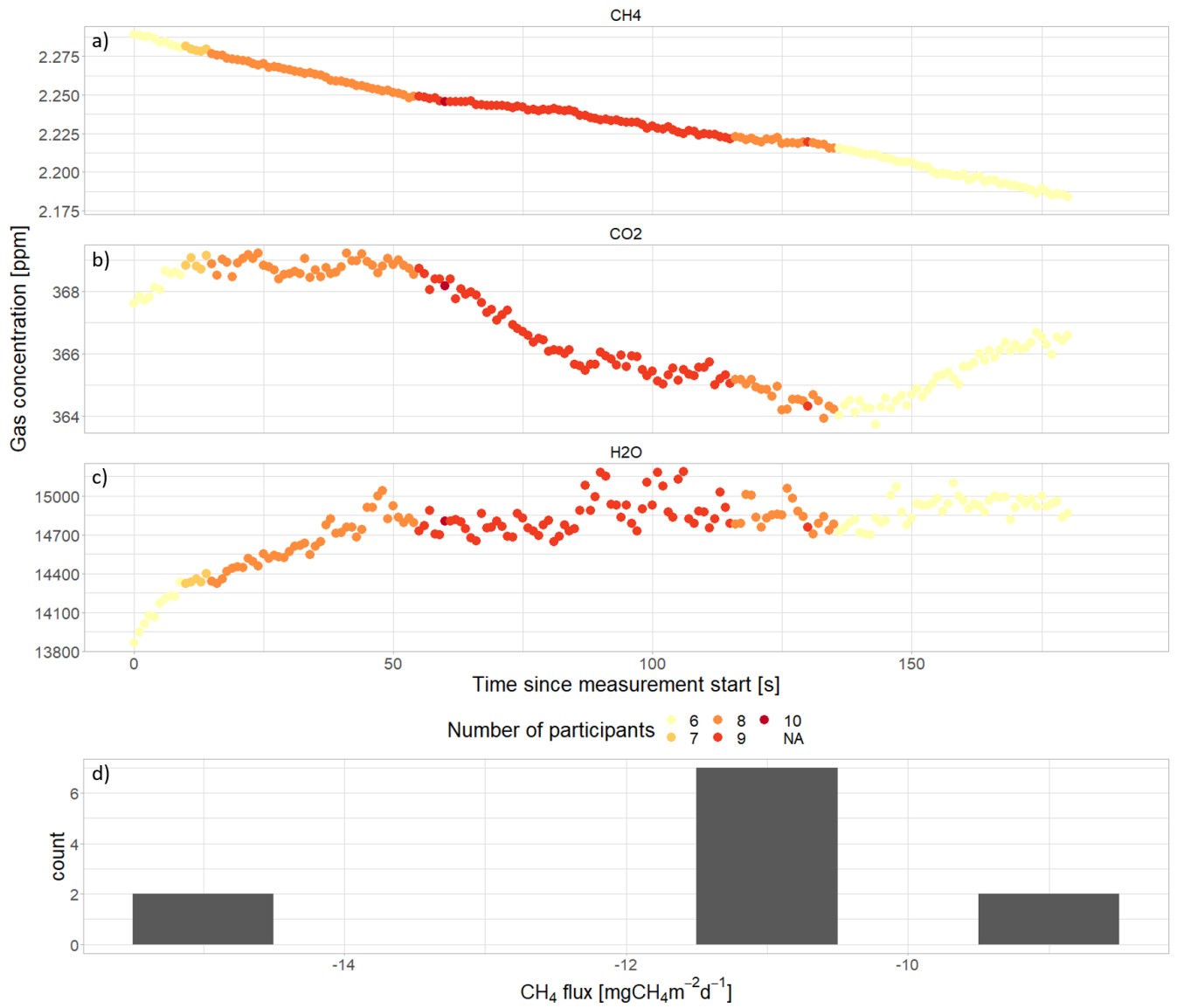

**Figure 17: Measurement example VQC3 of a small linear decrease in CH₄ concentrations over the time of the chamber closure.** Simultaneous measurements of CH₄ (a), CO₂ (b), and H₂O (c) concentrations over time during chamber closure with the colours of the data points indicating how many participants included the respective data point in the time period that they chose for flux calculation. Only the responses by participants who gave start and end times for flux calculation were considered for this figure. Histogram of CH₄ fluxes calculated based on the time period chosen for flux calculation by the participants (d).





### 3.2.8 No trend

In the visual QC exercise, we included one measurement example for which the $CH_4$ concentrations did not show a clear trend and varied only little over the time of the chamber closure (measurement ID VQC6, Figure 18). Most participants (69%) noticed the very small change in $CH_4$ concentrations over the whole measurement but they disagreed on whether the

concentration measurements represented a real flux. Half of them suspected a real emission that remained very small because of $CH_4$ production and oxidation cancelling each other out at a low water table and two more participants called it a "zero flux" where the uncertainty would likely exceed the flux magnitude. Some (39%) of the participants, however, explained the low change in $CH_4$ concentrations by air leaking from the chamber, two of whom related the leak to vegetation obstructing the chamber seal and one to lateral diffusion into the chamber from the surrounding area.

Some (19%) of the participants furthermore pointed out an inconsistent trend in the $CH_4$ concentrations which they related to a changing balance between $CH_4$ production and oxidation over time, noisy measurements due to a low precision of the gas analyser, or a bad chamber seal combined with wind disturbance. According to one participant the latter was supported by the fluctuations appearing in the concentrations of all three gases, while two other participants mentioned that the $CO_2$ concentrations looked linear, at least after 30 to 40 s into the measurement, indicating an intact chamber seal.

The majority of the participants (62%) decided to discard the measurement due to leakage from the chamber (38%), a changing trend in the $CH_4$ concentrations (44%), a too short measurement time (13%), or too higher initial concentrations of $CH_4$ and $CO_2$ (13%). While two of these participants manually set the $CH_4$ flux to zero, one participant pointed out that the concentration changes were too large to be below the precision of the instrument so that the measurement should not be accepted as a zero flux. Some (31%) of the participants kept the measurement assuming a small but nonetheless real $CH_4$ flux

and gave start and end times for flux calculation. Half of them discarded the beginning of the measurement as a period of initial equilibration, while the other half kept the entire measurement. The choice of different time periods for flux calculation by the participants resulted in a CV of 57% for this measurement example.

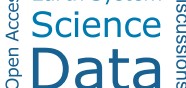

**Figure 18: Measurement example VQC6 with the CH4 concentrations showing little variation without a clear trend over the time of**
**the chamber closure. Simultaneous measurements of CH4 (a), CO2 (b), and H2O (c) concentrations over time during chamber closure**
**with the colours of the data points indicating how many participants included the respective data point in the time period that they**
**chose for flux calculation. Only the responses by participants who gave start and end times for flux calculation were considered for**
**this figure. Histogram of CH4 fluxes calculated based on the time period chosen for flux calculation by the participants including**
**the two participants who set the flux to zero (d).**


Earth System Science Data



### 3.2.9 Effect of measurement period and QC on overall flux estimate using a representative dataset

Some of the measurement classes that we defined to represent different patterns in $CH_4$ concentrations over the time of the chamber closure clearly occurred more frequently than others within the total of 788 measurements that we took at Siikaneva

bog (Table 1). The majority of the measurements (almost 60%) showed a linear increase in $CH_4$ concentrations in the chamber over the entire measurement. Nonlinear shapes with a decrease in slope over time were found in almost 20% of the measurements. Jumps in the $CH_4$ concentration either at the beginning or in the middle of the measurement were observed for 16% of the measurements while the remaining classes of an increasing slope over time, an inconsistent trend in $CH_4$ concentrations, linearly decreasing $CH_4$ concentrations over time, and no trend in $CH_4$ concentrations each made up less than

5% of the data set. Using the frequency of occurrence of each measurement class in the Siikaneva data set, we estimated an overall uncertainty of 28% introduced by different researchers discarding different numbers of measurements and an uncertainty of 17% due to different researchers choosing different parts of the same measurement for flux calculation.

## 4 Discussion

### 4.1 Implementation and development of Best Measurement Practices for chambers

Numerous attempts have been made towards standardizing flux measurements using static chambers by compiling and publishing guidelines on best measurement practices on chamber measurements (e.g. de Klein and Harvey, 2012; Fiedler et al., 2022). Our expert survey revealed that despite the existing guidelines, not all researchers are implementing the recommended measures for various reasons (Figure 4, Table 3). Reasons for not implementing certain chamber equipment or processing advice could be that some researchers might simply be unaware of the recent guidelines on chamber measurements

due to a lack of collaboration and a lack of suitable networking platforms. Only half of the survey participants are part of a flux monitoring network, such as FluxNet or AmeriFlux, most of which strongly focus on eddy covariance measurements. Other potential reasons for not implementing recommended measures include that existing equipment might not be suitable for adjustments, e.g. specific chamber sizes or shapes can make some measures inappropriate or impractical, financial constraints might not allow for improvements or new equipment, or there might be site-specific requirements on the chamber

design. The various scientific backgrounds of the researchers doing chamber measurements (Figure 2b) might further be contributing to the variety of measurement setups, calculation and QC procedures found in this survey as the educational training likely influences which aspects of the flux measurements, i.e. which chamber artefacts and environmental controls on $CH_4$ fluxes, are considered most important.

Some shortcomings in measurement setups can likely be related to an uncertainty around the effect of certain chamber

equipment on the measured $CH_4$ flux, leading to concerns that overcompensating for certain chamber artefacts can introduce new sources of error. While most researchers use fans to mix the air inside their chamber, some researchers mentioned that their chambers are small enough to not need a fan. This statement highlights that further research is needed to investigate the



strength of turbulence that is adequate for particular chamber dimensions to ensure proper mixing of the chamber air while preventing that additional gas is artificially released from the soil (Christiansen et al., 2011; Maier et al., 2022). Vents for

pressure equilibration are likely only used by half of the participants for the similar fear of causing a Venturi effect when wind passes over the vent outlet (Bain et al., 2005; Conen and Smith, 1998). Clear guidelines exist however on how to avoid the Venturi effect by adjusting the vent design (Xu et al., 2006). Furthermore, the two types of vents – the one that is open only during chamber placement and the one that remains open during the measurement – seem to be considered rather as alternatives for vent designs than as two measures that tackle different pressure-related chamber artefacts and that should therefore both

be applied simultaneously. The survey participants generally avoid the danger of overcompensating for a temperature increase inside the chamber and causing condensation (Fiedler et al., 2022) by using active cooling only in cases where it is required, e.g. because transparent chambers are used. Depending on the environmental conditions, opaque chambers could be used more often for insulation but blocking out the incoming radiation could reduce active $CH_4$ transport through plant aerenchyma thereby reducing the measured $CH_4$ emissions (Clough et al., 2020).

The ability to detect or correct for any remaining temperature and/or pressure differences between chamber and ambient air varies among the participants as not all of them record the temperature and/or pressure inside their chamber. Only two participants can account for temperature and/or pressure changes over the time of the chamber closure by individually correcting each concentration measurement as they document the chamber temperature and/or pressure at the same frequency as the gas concentrations. Changing temperature and pressure conditions inside the chamber might go unnoticed when using

only one temperature and pressure reading to correct the final flux estimate. Most notably, 17% of the survey participants do not measure the chamber temperature at all (Figure 4), which can lead to large uncertainties considering the large linear effect of temperature on the flux magnitude through the ideal gas law.

Similar to controlling the temperature inside the chamber the requirements for chamber dimensions, chamber insertion depths into the soil, and chamber seals strongly depend on the environmental conditions of the research site. Chamber

dimensions need to be adapted to the surface structure as well as to the vegetation height while allowing for flux detection within reasonable deployment times. As the required insertion depth of the chamber into the soil as well as the necessity of a gastight seal are low under water saturated conditions and at low soil porosities, the fact that one third of the survey participants did not use a collar or a seal on their chamber might be less problematic than it appears since many participants measure in wetlands or on open water. The chamber setup should nevertheless be tested for gas tightness before it is deployed in the field.

An increasing use of inline gas analyzers can loosen the requirements on chamber dimensions for $CH_4$ flux measurements due to shorter deployment times. In most researched environments $CH_4$ emissions are high enough so that the minimum detectable flux is reached quickly after the chamber closure. The high measurement frequency of inline gas analyzers therefore allows to significantly reduce the chamber deployment time, thereby decreasing the potential effect of chamber artefacts on the $CH_4$ flux estimates. Chambers can therefore be smaller as shorter measurement times reduce the potential

effects of leakage, lateral diffusion, temperature and pressure changes on the flux estimates. Currently, the majority of the survey participants (80%) use inline gas analyzers. Of the 26% of participants who manually sample the chamber air, two



participants (22%) keep their chamber closed for more than 40 min which is considered as too long by Rochette and Eriksen-Hamel (2008) but earlier guidelines allowed for up to 1 hour closure time (Holland et al., 1999). In addition to increasing the relevance of chamber artefacts due to longer closure times, manual sampling can obscure the influence of chamber artefacts

through the lower temporal resolution at which the gas concentrations inside the chamber are monitored. This limits the possibilities to still exclude measurement periods affected by chamber artefacts at the stage of flux processing and quality control.

The different variables measured alongside the fluxes might indicate that depending on their background and research questions the survey participants consider different variables as important in controlling $CH_4$ fluxes. The ancillary variables

also determine which additional information is available to the researchers to evaluate the quality of the $CH_4$ flux measurements. Almost all survey participants measured variables to characterize the soil, hydrological and meteorological conditions, covering most of the ancillary data suggested by Maier et al. (2022). The potential effects of the vegetation cover were however considered by less than one sixth of the participants only.

**Table 3: Recommendations for chamber design (adapted from Clough et al., 2020) (Roman), their implementations by researchers as derived from the expert survey and resulting issues (bold).**

| Design feature | Design objective | Minimum requirements | Site-specific issues | **Implementation by researchers** | Evolving issues | **Improvements** |
|---|---|---|---|---|---|---|
| AREA | Minimize error due to poor sealing and maximize area sampled. | A chamber area/perimeter ratio of ≥10 cm is recommended. (equates to a cylindrical chamber of ≥40 cm diameter). | Adaption needed if rocks or roots are present, or if required by research objectives. | **66 % of chambers have area/perimeter ratio of ≥ 10 cm. 75% of chambers with area/perimeter ratio < 10 cm are cylindrical.** | | **When using cylindrical chambers, make sure that they have a diameter of ≥40 cm.** |
| HEIGHT | Maximize flux detection and minimize perturbation of environmental variables. | Chamber height (cm) to deployment time (h) ratio should be ≥40 cm h$^{-1}$. | Chamber height should accommodate crop height. | **93% of measurement setups had chamber height to deployment time ratio of ≥40 cm h$^{-1}$. The two remaining setups had too long closure** | | **Inline gas analyzers allow for shorter deployment times, so lower chamber heights are acceptable as minimum detectable $CH_4$ flux is reached** |



| | | | | | | |
|---|---|---|---|---|---|---|
| | | | | times considering the relatively low chambers. | | after short deployment in most researched environments. |
| BASE DEPTH | Prevent below ground lateral gas transport, shading and ponding of water. | Ratio of insertion depth: to deployment time of ≥12 cm h⁻¹. Height above soil surface should be as close to the soil surface as practical (<5 cm). | **Required insertion depth is higher at lower soil porosity.** | **66% of the chamber setups involved a base (collar) inserted into the soil; 25% of the participants do aquatic measurements where no collar is needed, and 83% of the survey participants measure CH₄ fluxes from wetlands, where saturated soil conditions allow for low insertion depths.** | **Lateral diffusion effects on the concentration increase?** | **Inline gas analyzers allow for shorter deployment times as minimum detectable CH₄ flux is reached after short deployment in most researched environments, making shallower insertion depths are acceptable** |
| GASTIGHT SEAL | Prevent gas leaking between chamber and base. | A water trough or rubber/closed cell foam gasket. Gaskets should have low internal cross-sectional area and be compressible; appropriate fasteners are | **Appropriate sealing method depends on the environment. No sealing is required for aquatic measurements.** | **63% of the participants used a water trough or gasket to seal between chamber and soil or chamber base. 25% of the participants do aquatic measurement** | **Many chamber setups do not use a seal despite clear recommendation. Leaky seal effects on concentration increases?** | **Chamber setup should be tested for gas tightness.** |





| | | | | | | |
|---|---|---|---|---|---|---|
| | required with rubber gaskets. | | | **where no seal is required.** | | |
| VENT<br><br>i) while placing chamber on base<br><br><br><br><br>ii) during deployment | i) To prevent pressure disturbance while placing the chamber on the base.<br><br><br>ii) To prevent pressure gradients between the interior and exterior of the chambers during flux measurement and gas sampling. | i) Opening a vent or sampling port while placing the chamber is essential.<br><br>ii) Tube-type vents should be located close to the soil surface, or be designed to minimize wind effects. Appropriate vent dimensions (diameter and length) are dependent on expected wind speeds during deployment and should be adjusted accordingly. Chambers and their vents should be bench tested to ensure no Venturi effect occurs. Designs exist to | **Tube-type vents need to be adapted to expected wind speeds to avoid Venturi effect.** | **51% of chamber setups involve some kind of vent. Responses indicate that both types of vents are rarely used simultaneously.** | **Not used enough despite clear recommendation.** | **Both vent type i) and ii) should be applied. Danger of Venturi effect can be avoided following well-founded recommendation on the vent design.** |





| | | | | | | |
|---|---|---|---|---|---|---|
| | | overcome Venturi effects. | | | | |
| INSULATION | Prevent temperature gradients between the interior and exterior of the chambers. | Use reflective foil, foam, or polystyrene. Test effectiveness by comparing surface soil temperatures inside and outside the chambers | **Chamber needs to be transparent if CO₂ uptake through photosynthesis measured alongside the CH₄ flux.** | **3% of the participants insulate their chamber and 17% actively cool the chamber air.** | **Cooling of the headspace air relative to the ambient air which might lead to condensation inside the chamber or sampling tubes (Fiedler et al., 2022).** | **Use cooling system only if chamber cannot be insulated and/or if long chamber deployment periods are needed (Maier et al., 2022).** |
| HEADSPACE MIXING | Well-mixed headspace to ensure that representative sample is taken. | Active headspace mixing (e.g., fans) should not affect the diffusive flux. | Crop type and chamber height. | **80% of the chamber setups involve fans. 43% of the participants who do not use a fan stated that the air flow from circulation through a closed loop with the gas analyser sufficiently mixes the chamber air, particularly in small chambers.** | **Missing concrete knowledge on how much mixing is adequate for which chamber dimensions.** | Effects of mixing should be tested and reported on. There has been relatively little work performed on evaluating specific requirements for given chamber fan geometries and fan size–wind speed combinations. |

## 4.2 Considerations for data processing

While recommendations on the chamber setup and measurement routines are outlined in many places and well-founded by modelling and experimental studies, the steps for data processing, including data fitting methodologies and QC approaches

might also have a strong but less well understood effect on the calculated CH₄ fluxes.





A subjective assessment of different chamber effects led to the choice of different time periods for flux calculation and had the largest effect for measurements that showed a nonlinear increase in CH$_4$ concentrations (Table 1, Figure 6). Nonlinear measurements with a decreasing slope over time accounted for over 18% of our example data set, making it the second largest measurement class after linearly increasing CH$_4$ concentrations. Published studies confirm that this shape is

regularly observed in CH$_4$ concentrations from chamber measurements (e.g. Pirk et al., 2016). Biases produced by different processing approaches for these measurements therefore likely represent one of the largest sources of divergence between flux data sets processed by different researchers. To avoid any initial disturbance caused by the chamber placement from influencing the flux estimate, almost half of the survey participants generally exclude the beginning of each measurement from their flux calculation. For measurements with a nonlinear increase in CH$_4$ concentrations levelling off over time, this removes

the part of the measurement with the steepest increase in CH$_4$ concentrations resulting in lower flux estimates. In contrast, believing that the beginning of the measurement is least affected by chamber artifacts such as saturation, condensation, leakage, and temperature and pressure deviations from ambient conditions, about one third of the participants consider an exponential fit for their flux calculations, using the steepest, initial slope of the CH$_4$ concentrations as their flux estimate. This overall uncertainty on which part of a nonlinear measurement best represents the real flux also shows in the large range of flux

estimates derived for the nonlinear measurements with decreasing slope in the visual QC exercise (Figures 6, 11). The visual QC exercise furthermore revealed that in extreme cases where the trend in CH$_4$ emissions reverses from an increase to a decrease over the time of the measurement, interpretations of the concentration time series make the difference between quantifying net CH$_4$ emission or uptake from the same measurement (Figures 6, 16).

The survey participants were also divided on the question of keeping chamber measurements that similarly show a

nonlinear increase in CH$_4$ concentrations but with an increasing slope over time. This shape in CH$_4$ concentrations occurred less often in only 3% of the measurements in our example data set but was still reported surprisingly often also in other studies considering this behavior is not consistent with diffusion theory (Kutzbach et al., 2007). Accordingly, half of the participants who discarded the measurement supported their decision by stating that they cannot explain the observed shape in CH$_4$ concentrations and therefore cannot justify choosing a time period for flux calculation as this would largely affect the flux

estimate considering the nonlinearity. The flux estimates computed for the 65% of participants who kept the measurement and gave start and end times for flux calculation confirm that this concern was justified as the flux estimates differ by up to 78% depending on the time period chosen.

The informative value of the variance in flux estimates for measurements showing a nonlinear change in CH$_4$ concentrations might however be reduced as, for reasons of consistency, we used a linear fit for flux calculation even in cases

where participants suggested to use a nonlinear fit. A linear fit leads to a lower flux estimate than the initial slope of a nonlinear fit in the case of a nonlinear increase in CH$_4$ concentrations with a decreasing slope but to a higher flux estimate in cases where the slope in CH$_4$ concentrations increases over time. Depending on the shape of the concentration change as well as on the time period chosen for flux calculation and the responses of the other participants using linear as opposed to nonlinear fits might have thus either increased or reduced the variance of the flux estimates.



The examples showing a nonlinear increase in $CH_4$ concentrations furthermore revealed that the QC procedure is subjective in the way that it depends on previous experiences. For example, participants who had not encountered a certain shape in $CH_4$ concentrations in their own data sets before were more likely to discard the respective measurement example in the visual QC exercise. How frequently certain types of shapes in $CH_4$ concentrations appear in a data set in turn likely depends on the ecosystem type that is researched. Several participants therefore mentioned that they would like to see the entire data

set before deciding on keeping or discarding an individual measurement. Due to the limited number of example measurements, the visual QC exercise in our study might therefore not entirely represent the decisions that researchers would actually make when processing an entire data set.

       $CH_4$ ebullition is another phenomenon often encountered in wetlands (Green and Baird, 2013), affecting more than 16% of the measurements in our example data set, and that is handled differently by different researchers thereby introducing

additional uncertainty into data sets combining flux estimates from different researchers. Most importantly, the survey participants disagreed on whether ebullition events should be included in flux estimates from chamber measurements or if diffusive and ebullitive flux should be quantified separately, either by isolating periods of ebullitive and diffusive flux from one concentration time series or by separately measuring ebullition (e.g. Hoffmann et al., 2017), for example using bubble traps (e.g. Männistö et al., 2019). The visual QC of measurement examples showing jumps in the CH4 concentrations

furthermore revealed that there was some uncertainty surrounding the distinction between ebullition events and artefacts of the gas analyzer among the survey participants. The flux estimates of the 4% to 8% of the participants who would use a linear fit over an entire measurement containing episodic ebullition events to account for both diffusive and ebullitive $CH_4$ emissions were up to five times as high as the flux estimates from the participants considering the diffusive flux only (Figures 12, 13, 14). This highlights the importance of reporting whether ebullitive emission was included in a given flux data set.

The survey participants furthermore disagreed on whether the remaining part of a measurement after an ebullition event could still be used to quantify the diffusive flux. More than half of the survey participants (54%) kept the linear part of a measurement after an initial ebullition event for flux calculation while 38% of the participants discarded the entire measurement assuming that the high $CH_4$ concentrations in the chamber following the ebullition event would decrease the concentration gradient and thus decrease the $CH_4$ flux between soil and chamber headspace for the rest of the measurement

(Figure 13). This effect also influenced the range of flux estimates from a measurement with repeated ebullition events (Figure 14). Flux estimates from the 15% of participants using the shorter linear increase in $CH_4$ concentrations before the first ebullition event were three times as high as the flux estimates from the 19% of participants fitting the longer linear increase after the first ebullition event for their flux calculations.

       Similar to ebullition events, the conditions, and in particular the water table depths, under which net uptake of $CH_4$

can occur are debated within the flux community. Due to the linear development, flux estimates for a measurement showing a constant decrease in $CH_4$ concentrations over time differed less between the participants compared to nonlinear examples. However, more participants (54% compared to 12% to 31% for nonlinear measurement with constant trend) decided to discard the entire measurement because they do not expect a net uptake of $CH_4$ in a wetland despite the relatively low water table.





Another source of uncertainty lies in the identification and the handling of "zero fluxes". Two thirds of the survey participants discarded a measurement example in the visual QC exercise showing only very low variations in $CH_4$ concentrations without a clear trend over the time of the chamber closure. The other third of the participants made a flux estimate, 20% of whom set the flux to zero and 80% calculated a small positive flux. Only one participant remarked that the magnitude in $CH_4$ variations needs to be compared to the instrument precision to decide whether a measurement can be classified as a "zero flux". This approach gets closest to computing the minimum detectable flux (MDF) introduced by Christiansen et al. (2015) and refined by Nickerson (2016) to assess the quality of low flux estimates. Contrary to the approaches of most survey participants, Maier et al. (2022) recommend to not discard or set to zero fluxes estimates below the MDF.

Only one third of the survey participants assessed the uncertainty of their individual flux estimates which similar to the MDF allows for more profound quality-flagging of the data. One participant reports the differences between flux values resulting from different fit functions as uncertainty estimates. This allows to circumvent the decision between linear or nonlinear fit for flux calculation in cases where the reason for the nonlinearity in the $CH_4$ concentrations are unknown and it is thus not clear which function better represents the real flux thereby preventing strongly biased data sets. Assessing the uncertainty of flux estimates supports the request by Maier et al. (2022) that instead of being discarded all fluxes should be published in data bases together with quality flags or uncertainty estimates.

## 4.3 Remaining knowledge gaps and next steps

The assumptions and the uncertainty among the survey participants surrounding their selection of approaches for flux calculation and QC call for better diagnostics to unambiguously identify the processes that cause the observed patterns in the $CH_4$ concentrations. The survey participants suggested various reasons for a nonlinear behavior in $CH_4$ concentrations (Table 2) which strongly affected their decisions during flux calculation and QC and led to a large variation in the resulting flux estimates (Table 1, Figure 6). This poses the question of how to identify the actual reason for the observed nonlinearity which needs to be known in order to make an informed decision on when to redo or to discard a measurement and how to handle a measurement in flux calculation and QC in order to get an accurate estimate of the real $CH_4$ flux.

The occurrence for example of nonlinear changes in $CH_4$ concentrations or ebullition events might be influenced by the measurement setup but likely also depends on the environmental conditions. This indicates that in order to derive recommendations on how to best estimate the $CH_4$ flux from a given chamber measurement the environmental conditions under which certain phenomena such as different types of chamber artefacts like leakage and saturation or ebullition events can significantly affect the $CH_4$ concentrations in the chamber need to be identified. For example, as Pirk et al. (2016) showed, the unsubstantiated use of a nonlinear model for flux calculation can lead to an overestimation of $CH_4$ emissions and should therefore only be applied in cases where we can be sure that the nonlinear change in $CH_4$ concentrations was caused by a change in the gas concentration gradient. Improved diagnostics to identify the processes involved will therefore allow for more informed decisions on how to handle chamber measurements that show a nonlinear change in $CH_4$ concentrations and will





ultimately result in more accurate flux estimates. While most published scripts that acknowledge nonlinear changes in gas concentrations aim for a standardization of flux calculations by identifying the best fit based on objective criteria (e.g., Pedersen et al., 2010; Rheault et al., 2024), implementing such improved process diagnostics could make flux processing more adaptable

to individual field sites while maintaining a standardized calculation procedure.

Developing improved process diagnostics will also help to identify a standardized minimum set of variables that should be measured alongside the gas concentrations for additional quality control of the flux estimates. The visual QC exercise in this survey should be considered as an example of how researchers would handle a given data set rather than their own measurements as the additional information given in the visual QC exercise might not be entirely representative of the

additional variables that researchers would select to assess the quality of their flux estimates. For example, not all researchers might usually consider the behavior of the $CO_2$ and $H_2O$ concentrations when evaluating a $CH_4$ flux. In the visual QC exercise, however, most participants took the other gas concentrations into account as they were available while one participant mentioned that an unexpected pattern in the $H_2O$ concentrations alone would not make them discard a $CH_4$ flux measurement.

Similarly, the high number of measurements discarded in the visual QC exercise might be related to the survey

participants not having done the example measurements themselves. While the median percentage of measurements that the researchers discarded from their own data sets was 5%, the median share of measurements discarded in the visual QC exercise amounted to 19% when considering the number of occurrences of the different measurement classes in the Siikaneva data set. This might be related to the researchers not being able to redo the measurements in the visual QC data set as opposed to their own field measurements. Additionally, the Siikaneva data set, from which the weighting factors for the different measurement

classes included in the visual QC exercise were derived, might have contained more measurements that deviate from the expected continuous linear increase in $CH_4$ concentrations in the chamber than the data sets usually collected by the survey participants. As discussed earlier, this might be related both to the specific measurement setup used but also to the site-specific environmental conditions at Siikaneva bog.

**4.4 Implications: Effects of measurement variability and researcher variability and bias on methane fluxes estimated**
**using the chamber-based methods**

In this study we performed an expert survey to gain an overview of the approaches used by different researchers to measure, process, and quality control $CH_4$ fluxes using flux chambers. We analyzed the survey responses to identify the main discrepancies between the approaches used within the flux community as they might cause significant uncertainty when comparing or combining data sets collected and processed by different researchers. Synthesis datasets of chamber $CH_4$ fluxes

have been increasingly used to estimate $CH_4$ emissions for northern high latitude methane budgets (Kuhn et al., 2021; Treat et al., 2018). Generally, these synthesis datasets show consistent differences among different wetland classes, but additionally, high variability within each wetland class that is attributed to high spatial and temporal variability within the flux measurements that can be partly compensated for by using longer integration times (Treat et al., 2018) or by capturing spatial variability due to microtopography effects (Virkkala et al., 2024).





The results from this expert survey show that differences in methodology may be an additional factor contributing to high variability in methane fluxes found in synthesis datasets. For example, the handling of low positive and negative fluxes can significantly affect estimates of methane budgets particularly for high latitude and upland regions where low $CH_4$ emissions and/or uptake of $CH_4$ can be expected during large parts of the year. Discarding low fluxes or setting them to zero can therefore lead to a bias towards higher $CH_4$ emissions and potentially make the difference between a net annual uptake or a net emission

of $CH_4$ in low-flux regions.

    Still, the uncertainty caused by different researchers processing a representative data set as derived from our visual QC exercise seems small compared to estimates of the natural spatial and temporal variability in $CH_4$ fluxes. We estimated an overall variation between flux estimates caused by different researchers choosing different time periods of the same measurement for flux calculation of 17% and a variation in the percentages of measurements passing the visual QC of 28%.

These estimates are similar to the mean natural temporal variability of 19% but lower than the mean natural spatial variability in $CH_4$ fluxes of 88% calculated from autochamber measurements in five temperate and Arctic peatlands by Pirk et al. (2016). Pirk et al. (2016) similarly found that this natural spatial and temporal variability in $CH_4$ fluxes exceeds the difference between the fluxes estimated using different fit functions. However, it has to be noted that the uncertainty estimates derived in our study consider only the effect of differences in visual QC and do not account for different measurement setups and routines or

different fit functions used for flux calculation. Still, this discussion points towards the questions of where and when we introduce the largest error into our flux estimates – is it when we choose our measurement setup and processing approaches or do the location and the timing together with the spatial and temporal resolution of the measurements matter more? Answering this question will help to identify the most relevant starting points for improving the accuracy of flux estimates and help lower uncertainties for flux syntheses. In any case, the survey shows that our human decision making introduces uncertainties that

can obscure natural spatial and temporal variability in $CH_4$ fluxes. This might make it harder to identify relevant spatial and temporal environmental drivers of $CH_4$ fluxes, which is crucial for processed based modelling and model development.

**5 Conclusions**

With this expert survey, we aimed to get an overview of the measurement, data processing and QC techniques that are used within the flux community to estimate $CH_4$ fluxes from chamber measurements. By teasing out the main differences between

the approaches used, we identified major sources of uncertainty in syntheses of data sets processed by different researchers. From our findings we derived starting points for future research to address this uncertainty through refining and substantiating existing recommendations on the measurement setup as well as the data processing and QC approaches.

    From the survey responses, most but not all researchers are using a measurement procedure and all of the chamber equipment recommended to avoid chamber artefacts as much as possible. While well-founded guidelines on the measurement

setup exist in the literature, recommendations on flux calculation and QC approaches are less unambiguous. As a major source of potential bias in flux estimates, we identified widespread uncertainty among the researchers in the handling of chamber





measurements showing a pattern in $CH_4$ concentrations inside the chamber that deviates from the expected linear increase. The reasons presumed for the observed patterns in $CH_4$ concentrations were manifold which resulted in differing approaches on how to filter a data set as well as on choosing a time period for flux calculation within a measurement, causing uncertainties of 28 and 17%, respectively, when processing a representative data set. The choice of the time period strongly affected the resulting flux estimate especially for nonlinear changes in $CH_4$ concentrations. Another decision that significantly influenced the magnitude of flux values was whether to include ebullitive $CH_4$ emission in the flux estimates. Specific questions resulting from the expert survey are: (1) Which part of a nonlinear change in $CH_4$ concentrations best represents the actual $CH_4$ flux?; (2) Should ebullitive emissions be included in the $CH_4$ fluxes estimates from chamber measurements?; (3) Can a linear increase following an ebullition event still be used to calculate the diffusive flux?, (4) How should we deal with small changes in $CH_4$ concentrations over the course of a chamber measurement?; (5) Under which conditions can net uptake of $CH_4$ be expected?. Besides highlighting the crucial role of complete and details methods descriptions, all of these questions indicate that we need to better understand both the biogeochemical processes underlying the $CH_4$ fluxes as well as potential chamber artefacts.

Identifying the environmental conditions under which each of the processes can significantly affect the $CH_4$ concentrations inside the chamber as well as the environmental variables that need to be recorded alongside the gas concentrations in order to characterize these conditions will enable us to make more informed decisions on how to process chamber measurement in the future. Continuing to work towards a standardized measurement setup and measuring procedure for chamber measurements of $CH_4$ fluxes according to existing and well-founded recommendations will reduce the uncertainty between measurements collected by different researches. For data processing and QC, however, rather than being standardized, routines should account for the environmental conditions specific to each research site in order to avoid a bias in individual data sets. Establishing flux community networks for chamber measurements similar to the ones that already exist for the eddy covariance technique will foster a transparent exchange between researchers on measurement and data processing techniques. Such chamber flux networks could build on existing research infrastructures such as the LTER sites, the ICOS sites in Europe and NEON sites in North America. These measures will allow us to reduce the uncertainty in syntheses of data sets processed by different researchers which are urgently needed to estimate $CH_4$ emissions on large spatial scales. Making individual data sets more comparable and combinable forms a basis for addressing the need of data exchange that was recently acknowledged by the WMO as part of the "Global Greenhouse Gas Watch" (G3W) (WMO, 2023).

*Data availability.* The results of the expert survey described in this paper are available from PANGAEA: https://doi.org/10.1594/PANGAEA.971695 (Jentzsch et al., 2024b).



**Appendix**

a)

In the following you will find 12 examples of the change in $CH_4$ concentration over time (as well as the time series of $CO_2$ and $H_2O$ concentrations for additional information during one chamber closure of 5 min. The measurements were performed in Siikaneva bog (61°50'N, 24°12'E), Southern Finland at different measurement plots in summer 2021 and summer and fall 2022 using a manual flux chamber with a volume of 36 l equipped with a cooling system, two fans to mix the air inside the chamber, and a small opening for pressure equilibration. For the measurements, the chamber was placed on collars that were permanently installed in the ground. The gas concentrations inside the chamber were measured with an in-line gas analyzer in a closed sample loop at a measurement frequency of 1 Hz.

*Flux chamber*                    *Measurement plot*

We would like to know how you would handle the following examples for chamber $CH_4$ flux measurements in your data processing based on visual inspection of the change in gas concentrations over time.

Measurement information
- Siikaneva bog, Southern Finland, 61°50'N, 24°12'E
- Dominant vegetation: *Sphagnum magellanicum*, *S. rubellum*, *Eriophorum Vaginatum* is common
- Water table depth: -18 cm
- Date an Time: 2021/07/28 08:46 local time
- Transparent chamber
- Seal between chamber and collar: rubber skirt
- Gas analyzer: LI-COR LI-7810

b)
How do you explain the CH4 concentration change in the figure?

Would you discard this measurement?
☐ Yes, because…

☐ No, because…

If not, which part of the measurement would you consider for your flux estimate?
Please enter the start and end time in seconds after measurement start.

Start:

End:

**Figure A1: Example of the information provided (a) and the questions asked (b) for the visual QC of one of the 12 chamber**
**measurements under discussion.**

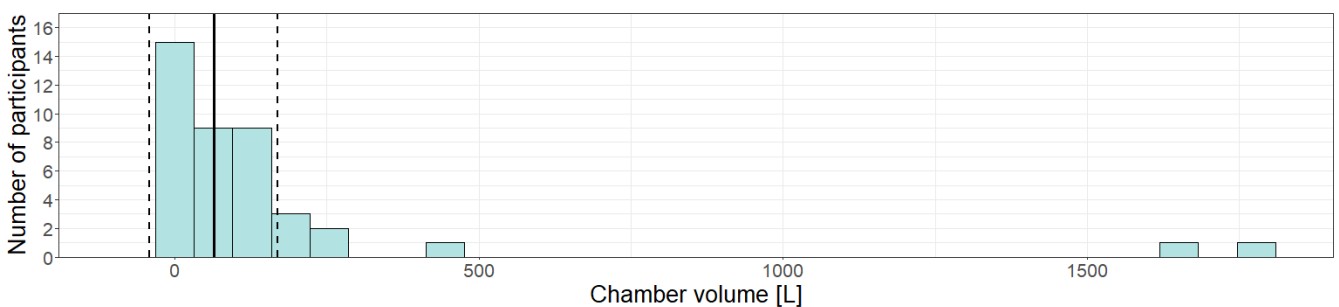

**Figure A2: Examples for different chamber setups in different environments. Automated chamber in a boreal forest (a), large**
**manual chamber with gas analyser insider the chamber (b), transparent manual chamber with in-line gas analyser and cooling unit**
**in a boreal fen (c), opaque manual chamber with a syringe for manual gas sampling and a tube for pressure equilibration (d), floating**
**chamber connected to in-line gas analyser and deployed from a boat for aquatic measurements (e).**



**Figure A3: Histogram of the chamber volumes. Black solid line: median, black dashed lines: IQR.**


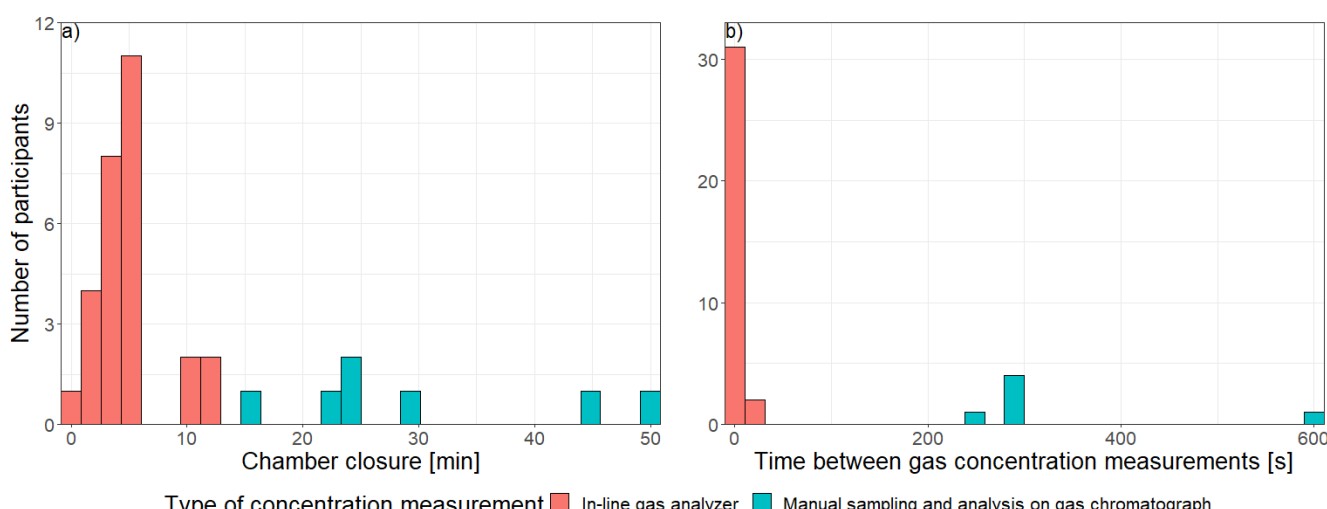

**Figure A4: Duration of chamber closure (a) and frequency at which the gas concentration inside the chamber is recorded as time between two measurements (b) by type of concentrations measurement.**

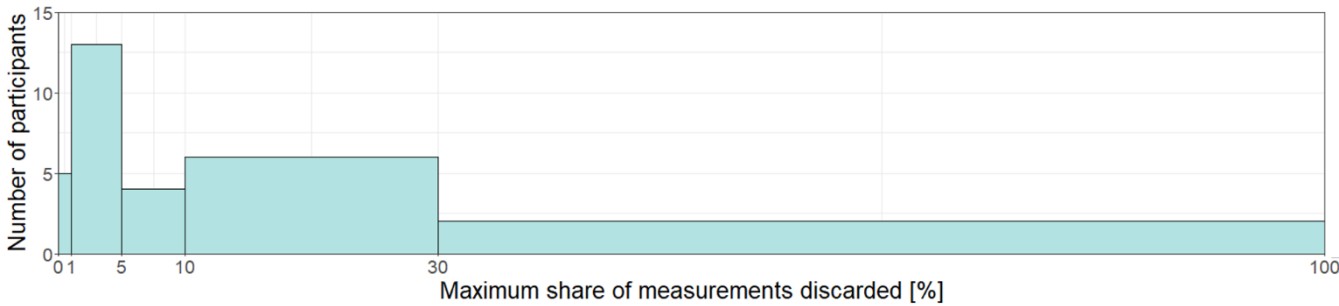

**Figure A5:  Maximum percentage of measurements that the survey participants usually discard from their own data sets.**

*Author contributions.* CCT, MF, and KJ conceived the project idea. KJ drafted the questions for the expert survey, set up the final online version of the survey, and collected the Siikaneva data set from which a subset of measurements was used as examples for the visual QC exercise. CCT, LvD, and MF reviewed the survey questions and tested the survey. KJ analysed

the survey responses and created the figures. The manuscript was written by KJ and commented on by all authors. CCT supervised the project.

*Competing interests.* The authors declare that they have no conflict of interest.

*Acknowledgements.* We thank Kathleen Hall, Katey Walter Anthony, Clayton Elder, Vasilii Petrenko for helpful discussions on methods that resulted in the conception of this project. We would like to thank Pamela Baur, Julia Boike, Jill Bubier, Jesper



Christiansen, Scott J. Davidson, Lona van Delden, Bo Elberling, Kathleen Hall, Paul Hanson, Nicholas Hasson, Liam Heffernan, Jacqueline Hung, Vytas Huth, Gerald Jurasinski, Sari Juutinen, Masako Kajiura, Evan Kane, Aino Korrensalo, Elisa Männistö, Nicholas Nickerson, Genevieve Noyce, Frans-Jan Parmentier, Matthias Peichl, Norbert Pirk, Maria Strack,

Eeva-Stiina Tuittila, Anna-Maria Virkkala, Carolina Voigt, Lei Wang as well as seven anonymous participants for completing the survey.

*Financial support.* The contribution of Katharina Jentzsch, Lona van Delden, and Claire C. Treat is part of the FluxWIN project, funded with a Starting Grant by the European Research Council (ERC) (ID 851181).

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
