# Peer review of "An expert survey on chamber measurement techniques and data handling procedures for methane fluxes"

_Earth System Science Data, 2024_

## Author Response (AR1)

**Response to RC1**

Dear Eyrún Gyða Gunnlaugsdóttir,

Thank you very much for your detailed comments which will greatly help us to improve the readability and accessibility of the information given in our manuscript.

In the following, please find our point by point responses (roman text) to your comments (*italicized*).

*It is a very interesting read and a great approach to the important task of unifying methods of chamber measurements for methane, an important addition to fill in a gap of knowledge. The article shows high quality work and data collection. It further on provides a starting point for future researchers to address and reduce the uncertainty of different types of measurement setups or processing of their data.*

*Easy to read for the most parts. Tended to be a bit text heavy where a table or a figure could have been presented in order to shorten the text. Which in turn would make it easier for the reader to find the results in a more concise manner.*

Thank you for this valuable comment. We agree that some parts of the description of the data set are unnecessarily lengthy. We have considered your specific suggestions given below for shortening the text and revised the entire text to make it more precise and easier to follow.

We have address all technical errors that you pointed out in the revised version of the manuscript.

*Chapter 3.1.3. This chapter would benefit from shortening the text. The schematic figure (4) at next page is very informative and already gives great info.*

We agree that the text is largely repeating the information that is already given in the figure. We have removed all information from the text that can also directly be seen from the figure. Instead, we have integrated the comparison of the measurement setups used by the survey participants with best chamber measurement practices from previous section 4.1 into section 3.1.3 (now 1.3.4).

*Figures 4 and 5 are excellent and give a great overview over this part of the results from the survey.*

*Chapter 3.1.4. This chapter would benefit from shortening the text and adding a descriptive figure/table that can be referred to.*

Thank you for this comment which supports a possible revision of the chapter that we were already considering. We have summarized the findings of this chapter in a flowchart showing the general process of flux calculation and quality control, complemented by the specific implementations by the survey participants and their frequencies of use (Figure 6 in

the revised manuscript). This allowed us to shorten the text and make the key results more easily accessible to the reader.

*Table 1. The table is taking up quite a lot of space, reaches over to the next page. The participants where all is NR/NA (participants 11, 12, 13 etc) can they be removed from the table and an explanation in the text why they were removed? Then the parts of the table that went to the next page can maybe be squeezed in to this page.*

Thank you for this valuable comment that has helped us to greatly improve the readability of this table (Tables A1 and A2 in the revised manuscript). We have removed the rows of the participants who did not give any responses in the visual quality control part of the survey as this information was not used in our uncertainty assessments. We have added a respective note to the table caption (lines 837, 838 in the revised manuscript).

*Also in the red boxes at the top of the table, some words are in two lines. Nonlinea – r. Insonsis – tent. The top row is taking up a lot of space and words are cut in the wrong places. The table otherwise is very informative and makes it easy to find the results in a short amount of time.*

We have corrected the hyphenation in the first row of the table (Tables A1 and A2 in the revised manuscript).

*Table 2. Very explanatory and easy to read for concrete results.*

*Chapters 3.2.2. – 3.2.8.*

*Figures show the results very well, easy to read. The figures do take up a lot of space. Can they be portrayed a bit smaller? With a less wide x axis, and even 4x3 subplots together in one page? If these figures could all be put into one or two pages, that would be great. Maybe the bigger figures could be in the appendix. Histogram's y-axis says "count", could that rather be participants?*

*These chapters are quite text heavy, they would all benefit from shortening the text and adding a descriptive figure/table that can be referred to.*

As both figures and text in chapters 3.2.2 – 3.2.8 took up a lot of space and contained a lot of detail, we have moved the entire chapters to the appendix of the manuscript (Text A1). We have instead summarized the key findings from these chapters in chapter 3.2 and Table 1 of the revised manuscript.

*Table 3. Very difficult to read the table. Could the table be horizontal instead of vertical? Then more text can fit there. The table might also fit better in the appendix. Text could be indented to the left.*

We have removed the table from the manuscript and have instead integrated the key information from the table into chapter 3.1.4 of the revised manuscript (former chapter 3.1.3).

**Response to RC2**

Dear Dr. Wiekenkamp,

Thank you very much for your detailed comments on our manuscript as well as on the data set itself. Your suggestions have greatly helped us to enhance the readability of the article and to clarify our study results. Defining a stronger focus of the article has also made the manuscript better fit for the scope of a "Visions" article in ESSD.

Based on your comments we have made two major adjustments to the manuscript:

1. In your comments, you rightly address the deviation of our manuscript format from a classical data set article. The question whether our manuscript fits the scope of ESSD has already been a point of discussion upon submission to the journal. The manuscript has therefore been transferred to "ESS Visions". We agree that in its previous state, our manuscript was somewhat of a mixture between a data set article and a research paper. In revising the manuscript, we have therefore taken full advantage of the opportunity given by an ESS Visions article to focus more strongly on our visions (Chapter 5 in the revised manuscript): how to close existing research gaps, i.e. approaches to avoid biases in chamber $CH_4$ flux data sets and to make them more comparable and combinable. We have therefore highlighted our recommendation of introducing a chamber flux network/ data platform with a standardized measurements protocol, standardized metadata requirements, and a data quality flagging system. Furthermore, we propose a further potential use of the data set underlying our survey as a standardized test and training data set for researchers to test their own flux algorithms and compare their results to others.

2. To clarify the focus of the manuscript and to enhance the readability of the article, we moved the sections where we discuss the detailed results of the visual quality control exercise to the supplement, which both reviewers found somewhat lengthy and text-heavy. This includes sections 3.2.2 to 3.2.8, including figures 7 to 18. In the main text of the revised manuscript we have summarized the key results from these chapters in chapter 3.2 and Table 1 to focus more strongly on the key uncertainties in the handling of the presented measurement scenarios. We have furthermore revised the previous detailed version of Table 1 by removing unnecessary rows (as suggested) and moved the table to the appendix of the manuscript (Tables A1, A2). We have added key results from the table in Table 1 of the revised manuscript and highlight the key variability estimates in chapter 3.2 (ll. 560-572)

In the following, please find our point by point responses (roman text) to your comments (*italicized*).

*I have read the manuscript "An expert survey on chamber measurement techniques for methane Fluxes ", written by Katharina Jentzsch and co-authors and considered for publication in ESSD. The article provides a new and unique dataset with expert survey results that evaluate the techniques scientists use to obtain, process and evaluate chamber-based methane fluxes at the field scale. This questionnaire does not only evaluate the specific chamber-based measurement method (system setup etc.), but also evaluates the techniques*

*that are used to process chamber measurements to ultimately obtain the reported methane fluxes. I think that, even though the number of participants is relatively low (as compared to general surveys), the amount of information obtained from the different questions is really nice and contain a lot of information. I think that the expert questions related to different measurement "scenarios" are very interesting and will probably really help the community to understand what discrepancies there are related to data processing and interpretation etc.*

*Although I really think the manuscript is well written, prepared and interesting, and I also see that the data set is valuable, the current manuscript format is (from my point of view) more similar to the format of a research paper and could be adjusted here and there (reformulated/ restructured, mainly in results, discussion and conclusion section) to really fully fit in ESSD as a data set article.*

Thank you for this comment. As stated above, we have revised the manuscript to focus more strongly on the conclusions for future research and actions that can be drawn from our study results in order to improve chamber $CH_4$ flux data sets to more clearly frame our results as an ESS Visions article. Furthermore, we have also expanded on the other potentially uses (and re-uses) of this dataset. (See chapter 5 in the revised manuscript).

*I also was wondering if one important key message is not just that it really could make sense to make raw or quality controlled (with quality id's) measurement data available in the future and to publish that for chamber-measurement articles? This would make it at least possible that people that want to use a large set of data from a lot of different areas can use their standardized processing and QAQC themselves to make sure that all the data is treated similarly.*

You are right – this is definitely something that we were thinking about but we seem to have ended up not writing it down explicitly. A data control flagging system was also suggested during discussions with other researchers. We have added this point to chapter 5 of the revised manuscript (ll. 695-698, ll. 727-731). This addition will help us further in taking full advantage of the opportunities given by an ESS Visions article.

*I overall enjoyed reading and have provided my (general and detailed) comments and suggestions below.*

**General comments:**

**1) Title of the manuscript:** *I am wondering if the wording "measurement techniques" covers everything mentioned in the manuscript. When I am thinking about the measurement technique, I generally consider mainly the instrumentation, measurement protocol and measurement principles. I, however, think that the survey also includes a lot of questions related to (1) the evaluation data quality control (QAQC) and the (2) data processing approaches. I am wondering if using an alternative wording, such as "data acquisition pipeline", "measurement framework" or perhaps "data acquisition framework" would help. Alternative, I can imagine that you could also use a second term in the title, such as "measurement techniques and data handling procedure". I think that an adjustment here*

*could potentially clarify that the survey was not limited to performing the measurement itself.*

Thank you for this very valuable comment. We have changed the title to "An expert survey on chamber measurement techniques and data handling procedures for methane fluxes".

***2) More focus on describing the dataset and explaining potential use case(s) of this dataset (applications/ limitations):*** *I think the paper has put a lot of effort in explaining what we can learn from the dataset and I can definitely understand that the survey creates an important step for awareness on measurement and workflow streamlining to obtain more comparable methane flux data (from chamber measurements). However, as far as I understand, the ESSD journal has a large focus on publishing these unique and important datasets itself – making it different from a general research publication (where one could also show the results of a survey). According to the ESSD website "detailed analyses as authors might report in a research article, remain outside of the scope of this data journal". Therefore, I would encourage the authors to be more concise with the results of the analysis (for example with the interpretation of the different graphs, especially from Figure 4 onwards) and put more focus on the description of the dataset itself.*

As mentioned before, your accurate observation that our manuscript falls between the article types within ESSD has already been a point of discussion with the editors. We have therefore revised the manuscript to further lean into having an ESS Visions article by adding more discussion about the implication and future applications of our study results (Chapter 5 of the revised manuscript).

***3) Recommendation on required methods information in research papers using chamber measurements:*** *As understood, the authors are often talking about making the gap smaller between approaches used in the community. At the same time, they state that details are often missing. Perhaps this publication is also a good place to make a statement on what information about measurement design, QAQC and processing is required in research paper manuscripts (to perhaps also have an uncertainty estimation when combining sets of measurements (based on your survey). I could imagine this to be valuable for the community.*

Thank you for this valuable comment. While we do not mention explicit information that should always be included in metadata reports, we have added our visions for future approaches to identifying and establishing a standardized set of metadata required for chamber measurements:

"Introducing a chamber flux network and data platform might speed adoption of a more standardized measurement protocol (although many recommended chamber components are widely adopted, Figure 4), improve metadata and ancillary measurements quality, spur development of a data quality flagging system that could foster a transparent exchange between researchers on measurement and data handling procedures and ultimately enhance the compatibility of individual flux data sets." (ll. 719-723 in the revised manuscript).

"[…] Additionally, model-derived metrics can be used in post-hoc quality control, as was demonstrated for the minimum detectible flux (MDF) metric by Nickerson (2016). Such

metrics will help identify a standardized set of required metadata on chamber setup and experimental design and ancillary measurements that should be takes alongside $CH_4$ fluxes in addition to the various variables currently recorded for the specific applications of the survey participants (Figure 5)." (ll. 768-772 in the revised manuscript).

***4) Representativeness:*** *I think one important element (that has to be clear in the manuscript and from the data set), where the authors need to elaborate on, is stating whether the people that are interviewed are a qualitatively good "representation" of the community. In other words – how would you judge the quality of the data set, based on the responses you had? What are the pros/ cons and what are the limitations of the 36 responses?*

Thank you for this valuable comment. We have added a chapter discussing the representativeness of the survey participants of the entire chamber flux community to the revised manuscript (ll. 605-622):
"**4.2 Representativeness of the survey respondents and questions**

From the variety of survey responses, it becomes clear that evaluating the representativeness of the respondents of the chamber flux community as a whole is challenging. One reason is that the chamber flux measurement community remains less organized than the eddy covariance flux measurement community and is more fluid, potentially because the barriers for entry are lower, i.e. the cost of analysis. We recruited the survey participants from different places of employment assuming that this would make them rather independent in their choice of measurement and data handling approaches. The main strength of the collected data set therefore lies in representing a large range of measurement and data handling practices; indeed, there were substantial deviations in workflows within the part of the chamber flux community represented in this survey (Figures 4, 6). However, we did not reach all researchers using chamber fluxes with our survey; we likely underrepresented those working in agricultural ecosystems, disturbed sites, and tropical ecosystems. Overall, participants who had not encountered a certain shape in $CH_4$ concentrations in their own data sets before were more likely to discard the respective measurement example (Table 1, Table A2). For example, the measurement showing decreasing $CH_4$ concentrations over time was discarded by 50% of the current participants (Table 1), many of whom focus on wetland ecosystems (Figure 3), but is more likely to occur in well-drained agricultural soils (Mosier et al., 1997). Thus, the background of the survey participants might have affected the outcome of the visual QC exercise with a bias towards expected (higher) fluxes.

Additionally, the question of number of survey participants is always a concern. While the number of researchers contacted (n=46) and the final maximum of 36 respondents might seem relatively low for a community survey, we estimate that this still represents a considerable extent of an estimated total number of several hundred chamber flux experts world-wide."

***5) Choice for type of question:*** *In your questionnaire, you have made clear choices for using specific types of questions. Surely, the type of question also affects the information you can*

*obtain          from          the          survey          (e.g. https://www.soscisurvey.de/help/doku.php/en:create:questiontypes ). It would be great if the authors could elaborate on the choice for the specific question-format they have chosen. The authors have, for example, not used any Likert scale question and have used yes/no questions a lot. Consider also adding some information about these question choices into the manuscript. If the question design has specific pros/ cons, limitations, this could also be discussed in the manuscript.*

We highly appreciate your comments on the appropriate handling of survey data and the required information. Thank you for providing the link to this very useful overview of question types. We have added a paragraph explaining our choice of question types to section 4.2 of the revised manuscript (ll. 620-629):

"[…] the question of number of survey participants is always a concern. While the number of researchers contacted (n=46) and the final maximum of 36 respondents might seem relatively low for a community survey, we estimate that this still represents a considerable extent of an estimated total number of several hundred chamber flux experts world-wide. Time is always a factor in voluntary survey participation; therefore, it was important to streamline questions to incentivize survey completion. In offering a diversity of question types, we attempted a balance between making the responses comparable and categorizable among the participants while still obtaining detailed information on their reasoning for the use of specific measurement and data handling techniques. The limited number of survey participants required a low number of possible responses in choice questions to allow for a meaningful statistical interpretation of the survey results; therefore, we used yes/no answers rather than scales of agreement. Yes/No questions further allowed us to draw conclusions on the prevalence of the implementation of recommended best measurement practices among the survey participants."

**6) Methods – Use of chamber measurement data:** *In this questionnaire and data set, measurement from your field campaigns were used for the evaluation of flux processing were used. The authors have explained that they have used a lot of different types of measurement scenarios that can occur. Perhaps the authors can elaborate on (1) the representativeness of the 12 measurement "scenarios" that were chosen (any specific measurement case missing/ not available/ not occurring in your ecosystem/ not present for your measurement system) and (2) the representativeness of this study site for the evaluation of the fluxes and their uncertainty. I can, for example, imagine that the quantitative numbers (and absolute fluxes) would be very different for tropical wetlands.*

We have added a chapter discussing the representativeness of the Siikaneva data set to the revised version of our manuscript (ll. 630-661):

"**4.3 Assumptions in the flux calculations: site and researcher differences**

Our estimates of researcher variability in flux data sets, derived from the visual QC exercise, strongly depended on the underlying reference data set collected at Siikaneva Bog (Table A1). Both natural processes and chamber-induced artefacts occur and their prevalence

depends on both the environmental conditions of the research site as well as on the chamber design and measurement setup. Most measurements in the Siikaneva data set (~60%) showed the linear increase in $CH_4$ concentrations that is expected for an undisturbed measurement at a wetland site. However, a nonlinear, weakening increase in $CH_4$ concentrations was also represented by a rather high share of measurements (18%) and is also regularly observed at other sites (e.g. Pirk et al., 2016). The survey responses confirm that it is often unclear whether this shape is caused by an initial disturbance of the measurement or by $CH_4$ saturation of the chamber headspace over time (Table 1). Furthermore, this lack of process-understanding shows through in the high variance associated with the non-linear fluxes (Figure 7, Table A2). An initial disturbance, i.e. ebullition caused by the chamber placement, was a common explanation (Table 1) and might have occurred more frequently in the Siikaneva data set, than other sites, as roughly 60% of the measurements were obtained from vegetation removal plots. The removal of vascular plants and of the *Sphagnum* moss layer might have reduced both plant-mediated $CH_4$ transport and $CH_4$ oxidation, resulting in higher $CH_4$ concentrations in the pore water and thus increasing the probability of ebullition events (Jentzsch et al., 2024a). While $CH_4$ ebullition is a natural phenomenon often encountered in wetlands (Green and Baird, 2013), the increased probability of both natural and anthropogenically induced ebullition due to vegetation removal might have contributed to the high share of measurements (16%) showing abrupt jumps in $CH_4$ concentrations in the Siikaneva data set.

Although some measurement scenarios included in the visual QC exercise are relatively uncommon, it is still important to evaluate how these scenarios would be handled by different researchers as they showed large sources of disagreement (Table A2). Many survey participants stated that the nonlinear increase in $CH_4$ concentrations with an increasing slope over time was unexpected. However, this shape was reported surprisingly often in other studies and occurred during several of our measurements (Table A1). Overall, this behavior of $CH_4$ concentrations in the chamber headspace is not consistent with diffusion theory (Kutzbach et al., 2007), indicating the influence of other processes. Similarly, both low changes in $CH_4$ concentrations without a clear trend and a decrease in $CH_4$ concentrations over time occurred infrequently in the Siikaneva data set (<1% of measurements) but were scenarios with high variability in the calculated fluxes (Table A2). Still, small fluxes might be expected at higher and drier wetland microtopographical features (e.g., Laine et al., 2007), while low, close-to-zero fluxes or $CH_4$ uptake are more commonly observed at upland sites (Virkkala et al., 2024; Voigt et al., 2023).

Overall, the Siikaneva data set might have contained more non-linear measurements than data collected by the survey participants due to the selected experimental setup as well as site-specific environmental conditions. This theory is difficult to test as this information is not often available for other sites but might be a reason for the high discard rate in the visual QC exercise."

**7) Results, Discussion, Conclusion:** *I think the results should be more focused on giving an overview over the dataset and what the dataset shows/describes (see also comment 2). Especially the measurement example results (including all figures related to this) make the*

*paper quite lengthy in the end. I suggest shortening here (one could potentially put some of the result figures in the appendix). I would especially consider shortening from Figure 6 and section 3.1.4 onwards.*

We agree that the description/ interpretation of the individual measurement examples in the visual quality control exercise is quite lengthy and this was also pointed out by the other referee. We have therefore moved the sections 3.2.2 to 3.2.8 including figures 7 to 18 to the appendix (Text A1). We have summarized the key findings from these chapters in chapter 3.2 and Table 1 of the revised manuscript.

*I am also wondering if, instead of having a very detailed discussion section is appropriate in such dataset paper. Instead of following the typical research paper setup and coming up with a full discussion about the results, I would suggest to stick more to demonstrating what the data set is about, what it can be used for, messages it can convey, etc. I could imagine that a separate paper in the form of an opinion paper or the like could be made to create new guidelines/ setting up certain rules and refer to the current data set.*

As explained above, instead of turning the manuscript into a clear data paper, we have decided to lean more strongly into the opportunities of having an ESS Visions article by expanding on future implications of our study results, as suggested here with the possible opinion paper. We have therefore replaced the previous discussion chapter by a chapter focusing on our vision for future improvements of chamber $CH_4$ flux measurements and data sets (Chapter 5 of the revised manuscript).

*Also consider all the above-mentioned comments for the conclusion section. I think that this section is also written in the shape of a typical research paper. I would consider rewriting and renaming here to more focus on the data set and what one could use it for.*

We have revised the conclusion section to highlight the researcher variability identified in our study and its potential implications for data synthesis and modelling efforts.

**Detailed comments:**

1. **Abstract, Line 11-12, 2nd and 3rd sentence**: *"To estimate methane emissions … methane fluxes within the flux community." I could imagine that the transfer to chamber measurements from the 2nd to 3rd sentence is a little abrupt here. While reading I assumed that the authors first spoke about different methane flux measurement methods (e.g., chamber measurement regional airborne mass balance campaigns, tower-based eddy covariance, airborne eddy-covariance, gas column concentration inversion), but the second sentence here is directly directed to chamber measurements. I suggest considering either a direct focus on chamber measurements, or making a clear transfer from all methods to chamber-based fluxes.*

   Thank you for this comment. We have clarified by adding a sentence in between, introducing chamber measurements as one type of flux measurement techniques:

"One common method for obtaining in-situ methane flux measurements are flux chambers. " (ll. 13,14 in the revised manuscript).

2. ***Abstract, line 14 – 15****: "Existing guidelines on … within the flux community." I suggest to be more precise here when defining "flux community". I think that this community is beyond people that perform chamber measurements and therefore the second part of the sentence might be not very accurate. Also, I would suggest redefining methods (as suggested for the title), as you are not talking about measurement methods to measure methane fluxes, but really about differences in measurement setup, processing, etc. for flux-chamber measurements.*

We have clarified by changing the respective sentences to "Existing guidelines on chamber measurements promote more standardized measurement and data processing techniques but to our knowledge, so far, no study has investigated which methods are actually used within the chamber flux community. Therefore, we aimed to identify the key discrepancies between the measurement and data handling procedures implemented for chamber methane fluxes by different researchers." (ll. 16-19 of the revised manuscript)

We have clarified that we are only considering chamber-based measurements and the researchers using flux chambers in our study throughout the manuscript by using the term "chamber flux community". We have furthermore specified the term "methods" throughout the manuscript.

3. ***Abstract, Line 18-19****: "We conducted an expert survey … quality control of data." I would be more precise here as well. Specifically, the segment "to collect information on chamber-based methane flux measurements" could express more precise what you collected (e.g., to collect information on how scientists (?) conduct chamber-based methane flux measurements, etc.…).*

We have rewritten this sentence to clarify that we are investigating how different scientists conduct chamber measurements and handle the resulting data: "We conducted an expert survey to collect information on why, where, and how scientists conduct chamber-based methane flux measurements and how they handle the resulting data." (ll. 20,21 of the revised manuscript)

4. ***Abstract, Line 21****: "… with most measurement times falling between 2 and 5 minutes." Consider being more specific here with "measurement times" - maybe more clearly mention here that you're talking about the total measurement time (right?) to obtain (high frequency, > 1 Hz) methane concentrations (and total closure time of the chamber?).*

We have rephrased the sentence and used the suggested term "closure time of the chamber" instead of "measurement time": "We received 36 responses from researchers in North America, Europe, and Asia which revealed that 80% of

respondents have adopted multi-gas analyzers to obtain high-frequency (< 1 Hz) methane concentration measurements over a total chamber closure time of typically between 2 and 5 minutes." (ll. 21-23 of the revised manuscript).

5. ***Abstract, Line 25***: *"…on processes" Not clear if you are talking about the fact that scientists process the data differently or that they're thinking about different processes being held responsible (I assume the first). Please consider rephrasing.*

   We actually meant the latter. We have rephrased the sentence to clarify: "The responses showed broad disagreement among the experts concerning the processes that they consider responsible for nonlinear methane concentration increases." (ll. 26-28 in the revised manuscript).

***Abstract***: *I recommend the authors to add a direct link to the data within the abstract. This makes it easy for everyone to access your data directly, even if they only read the abstract.*

Thank you for this suggestion – we have added the link at the end of the abstract: "The survey results as well as the questionnaire are publicly available at https://doi.org/10.1594/PANGAEA.971695 (Jentzsch et al., 2024b)." (ll. 35,36 of the revised manuscript).

6. ***Introduction, Line 45 - 48***: *"achieved …. on the microscale." Here, the description of gas measurements has a focus on the soil. I suggest to consider plants here as well in the description…*

   We have revised the sentence to account for the role of plants in gas exchanges: "When applying the closed-chamber technique, a chamber is placed on top of the soil and the change in gas concentrations in the chamber headspace is monitored over time to estimate the exchange of $CH_4$ between soil, plants, and atmosphere on the microscale (e.g. Livingston and Hutchinson, 1995)." (ll. 50-52 in the revised manuscript).

7. ***Introduction, Line 74-75***: *"establish a more standardized protocol for measurements". Would be great to mention here if these protocols focus on measurement design and equipment only or if they also discuss the data processing and quality control as well.*

   We have specified the focus of the individual studies cited: "These best-practice guidelines for chamber measurements summarize recommendations on chamber designs (e.g., Clough et al., 2020) as well as on the entire workflow from measurement to data processing and quality control (e.g., de Klein and Harvey, 2012; Fiedler et al., 2022; Maier et al., 2022)." (ll. 82–84 in the revised manuscript).

8. ***Introduction, Line 90 – 91***: *"This study aims … data sets." This comment relates to the general statement (2) that this paper is should focus on presenting the dataset. I would therefore focus here more on the fact that you are presenting this new dataset and what this new dataset is envisioned to provide help with/ answer questions on for the community.*

We have rephrased the study aims to highlight the nature of the manuscript as an ESS Visions article: "This study aims to derive starting points for improving the usability of chamber $CH_4$ flux data sets for large scale synthesis studies through reducing the discrepancies between measurement and data handling approaches used within the chamber flux community as identified from an expert survey. Our objectives were to (1) provide an overview of the chamber designs, measurement setups and routines, flux calculation and QC approaches that are currently used by scientists to quantify $CH_4$ fluxes; (2) estimate the variability that is introduced into $CH_4$ flux data sets by the variety of data handling approaches when a representative data set of chamber measurements is processed by different researchers. Our study raises awareness for differences in chamber methods used within the flux community – a potentially considerable but often neglected source of error in synthesis studies that combine flux data sets collected and processed by different researchers. Through identifying major sources of uncertainty resulting from the variety of measurement, calculation, and QC approaches used within the chamber flux community, we derive starting points for eliminating such error sources and rendering individual flux data sets more comparable and combinable and thus better suited for larger scale synthesis studies." (ll. 97-107 of the revised manuscript).

9. ***Introduction, Line 76 – 81:*** *"While guidelines … used by individual researcher". "At the same time … data set highly uncertain". Consider rephrasing to really make clear what's the problem statement and what is currently missing. I think the statement in the last sentence of this block has much more weight (and sounds potentially more serious) as compared to the first. The first sentence sounds as if there are clear guidelines, but they are not followed (not sure if there's proof for that – might also sound a bit as if people are not well-behaving). The last sentence on the other side sounds rather different – as if there are not yet a complete community-based streamlined set of rules on processing and quality control. Alternative, as mentioned before in the beginning of the review, the interpretation by different scientists could still be different (with motivations specified). However, if the full dataset is available, streamlined flux calculation for larger data sets could still become more feasible/ realistic.*

This is a very valuable comment. We have revised the section to stress the actual problem that methods are often not documented sufficiently instead of emphasizing what we guess is the case or learned from personal exchange: "While guidelines outlining best measurement practices for chamber measurements provide a well-founded summary of methods recommended to collect high-quality flux data, chamber-based flux data sets are often lacking detailed metadata reporting on chamber design, flux calculation and QC methods. This introduces substantial uncertainty to comprehensive comparisons of chamber-based data." (ll. 84-88 in the revised manuscript).

10. ***Methods, Line 103:*** *"Experts were required … of measurements." I suggest to be more specific here and mention the need for expertise with chamber-measurement for methane fluxes, which I assume was your prerequisite.*

Yes, the word "expertise" is missing here. We have rephrased the sentence accordingly: "Experts were required to have a minimum expertise of one field season of chamber measurements of $CH_4$ fluxes." (l. 113 in the revised manuscript).

11. **Methods, Line 106:** *"Altogether, 46 experts were contacted via email". It would be good to put this number here in perspective. Considering the chamber-measurement flux community, is this a representative group?*

We have elaborated on our choice of flux experts that we contacted and put their number into perspective considering the entire chamber-measurement flux community of maybe several hundred researchers: "While the number of researchers contacted (n=46) and the final maximum of 36 respondents might seem relatively low for a community survey, we estimate that this still represents a considerable extent of an estimated total number of several hundred chamber flux experts world-wide." (ll. 620-622 of the revised manuscript)

12. **Methods:** *I think it would also be good to check if, and to mention how your survey and the storage of the provided data set on Pangaea is conform with the EU data protection regulation (just to be on the safe side).*

Thank you for this very important suggestion. The survey itself was officially checked in terms of data protection regulations before application. We have added the following statement to the manuscript: "The survey has been legally checked by a data protection officer to comply with the EU data protection regulation and involved a privacy policy statement explaining the use and processing of the collected data that needed to be approved by every survey participant prior to participation." (ll. 121-124 of the revised manuscript).

13. **Methods, Line 112 – 115:** *"This part of the survey contained 40 questions …image file upload." I was not sure how to get to the 40 questions (I got to 39) and would recommend to consider rewriting to make it easier to read (and easier to understand).*

We had included the image file upload as one of the questions. We have rewritten the sentences to make them easier to understand: "In the first, informative part of the survey, we gathered information on the measurement, data processing and QC approaches that the participants use for their own chamber measurements. For this part of the survey we chose a combination of 20 choice questions (simple and multiple selection including seven yes/no questions), all of which offered to elaborate the selection(s) in a short free text comment, and 19 text entry questions. For a visual overview of the variety of measurement setups used, we asked the survey participants to upload a photo of their chamber system." (ll. 124-128 in the revised survey).

14. **Methods, Survey part 2:** *Last sentence of the paragraph – "We asked them … of the response". Here it would be maybe good to explain more exact what you asked (start and end seconds). I think this makes it more complete and connects it better to the section where the survey results are shown.*

We have made this sentence more explicit by mentioning that we asked for the seconds after chamber closure at which the participants would start and end the time interval of the chamber measurement that they would consider for flux calculation: "If they decided to keep the measurement, we asked them to select the part of the measurement that they would use to calculate the $CH_4$ flux by submitting the start and end times of this period in seconds after chamber closure." (ll. 155-157 in the revised manuscript).

15. **Section 2.3.2, Line 178 – 180:** *"For reasons of consistency … were given by the participants". In these cases, I am wondering how much change a non-linear fit would have given. If the difference in flux would be very large due to the use of a different model, would it not be important to include this? Especially since this could affect (and increase) the uncertainty of the measurement.*

This is correct – including nonlinear fits could strongly affect the estimated uncertainty. We have seriously considered your suggestion of performing nonlinear instead of linear fits when suggested by the respondents. However, we have in the end decided to keep the linear fits only because often the type of nonlinear fit was not specified and because we could not be sure that all participants who would rather use a linear fit actually considered that this was an option in this visual quality control exercise. We have added a paragraph discussing this decision to the revised version of the manuscript: "If respondents did their own flux calculations, this would allow for non-linear fitting methods, which we did not use in our exercise despite being occasionally suggested by a participant (7% of responses). While our fitting and calculation approach may have been overly simplistic, post-hoc assumptions of how many participants would have used a non-linear fit and the different fitting options (such as exponential, quadratic, or logarithmic functions) would introduce substantial additional uncertainty into our estimates of researcher variability. Reproducing the calculation approaches of every respondent would have required additional, very detailed information from the survey participants, likely reducing the number of completed surveys and making our uncertainty estimates less representative of the entire chamber flux community. However, this type of exercise might be worth undertaking in the future." (ll. 670-677 in the revised manuscript).

16. **Results, Figure 2b –** *Maybe it is good to mention in the caption of the figure that participants gave multiple answers regarding their scientific background (causing the total number of responses to be above 36). I could imagine this could also help in other figures.*

We have clarified this in the captions of figures 1, 2, 4, and 5 of the revised manuscript.

17. **Results, Figure 3:** *I did not find this figure fully intuitive from the start (some of the stand-alone words only made sense after reading the caption of the figure). I also am not 100% sure if this figure is needed, also after having already Figure 2.*

We have reconsidered the significance of this figure while revising the manuscript. We consider the ecosystem types where chamber measurements are applied as interesting because they might affect the suitability of specific measurement setups as well as the occurrence of certain measurement classes (development of $CH_4$ concentrations over the time of the chamber closure) and thus the relevance and applicability of certain processing procedures. We have therefore decided to keep the figure, but added short headings ("Climatic zones", "Ecosystem types", "Wetland types") to the three parts of the figure to make its content more readily accessible to the reader.

18. *Results, Line 238 – feel like this block of text might need a new heading – e.g. research goals or something the like.*

Yes, we have made this paragraph a separate section on the research goals of the participants that has its own heading (Chapter "3.1.3 Research goals" ll. 266-280 in the revised manuscript).

19. *Results, Figure 4: The total % of the two different sampling techniques does not add up to 100% (26 and 80). Is this again related to the fact that researchers can choose multiple options? In that case, it would be good to mention this in the caption of the figure.*

Yes, several participants mentioned that they use both manual sampling and in-line gas analyzers. We have clarified that the researchers could choose multiple options in the figure caption (ll. 429-431 of the revised manuscript).

20. *Table 3 and 1: I think that Table 3 and 1 are not very easy to understand/follow and are both very lengthy in their current form. Considering the fact that this is a data set paper, I suggest that these tables could be simplified with some key messages, adjusted and added to the appendix, or simply be left out (and instead write some important statements as text). The dataset will be there to give the people that are interested detailed information.*

We have simplified Table 1 and moved it into the appendix (Tables A1 and A2 in the revised manuscript). We have furthermore removed Table 3 and instead integrated its key statements into section 3.1.4 of the revised manuscript.

**Dataset specific comments:**

*I have looked at the dataset on PANGAEA. Based on the dataset presented on the webpage https://doi.pangaea.de/10.1594/PANGAEA.971695. The dataset is generally in a good shape, but the documentation could be improved. Suggestions that can help to improve the dataset are listed below:*

We have revised the dataset for PANGAEA based on your comments and resubmitted the revised files. The PANGAEA webpage will be updated within the following weeks.

1. **Name of file(s):** I would improve the title of the excel file – A name that directly links to the Pangaea page and/ or the publication would be very nice.

   We have revised the file names to include authors, title, year of publication, and file content.

2. *Personal information: Although the survey results as presented do not contain names etc. and are anonymous, the images of the system setup could potentially relate the science group/ scientist to the answers provided (e.g., thinking of google search that has an image option – specifically if these images are online). Also, the images quickly could provide information about the more specific research area and relate this to PIs/ research groups (people might know each other's field work areas, equipment etc.). I could imagine that a storage of the different measurement systems separately (e.g., as .png/ .jpeg) could make the results less personally sensitive (thinking about data-privacy), while keeping the information available for others to use.*

   Thank you for pointing this out. While we have asked every survey participant for permission to upload their photos, we have now uploaded the photos in folder separate from the remaining survey responses.

3. *Consideration(s) file format: While opening the survey in excel, the response of some questions with Boolean arrays (e.g.: TRUE, FALSE) shifted to the local pc language. Choosing a different data format – such as .csv or .txt could make sure that things, such as change of a date-formats (which happen in excel), do not happen. This also incorporates that you have taken care more of the "I" in FAIR data – it's probably more interoperable in a different data format. I could image that creating separate files for the different tabs in the excel file would make sense (e.g.: Survey data, VQC data, demographics). When considering a change in file format, it is also nice to make it easy to import into R or Python for example. To simplify the work, the photos that the survey refers to can be stored separately as figures.*

   Thank you for this advice. We have changed the file format to .csv.

4. *Metadata: In several cases, information about units is not provided (for example in VQC data). It is important to state variable name, description of the variable name, unit and any other important information for such files to be fully used for further research. This includes an explanation on what NA means. Examples of such tables can be found in a paper by Loritz et al., 2024 (https://essd.copernicus.org/articles/16/5625/2024/essd-16-5625-2024.html) for example. This metadata should be added to the Pangaea data repo to make it easy for people to use the data once they download it. I generally also would encourage a short description document that accompanies the dataset (with information about variables, link to article, data license etc.), but this is just a personal recommendation.*

   We will have added a .pdf file giving an overview and explanations of the provided files and the variables and their units used in them.

5. **Link to paper and survey:** At a later stage, the link to the paper on the Pangaea page would be important to directly switch back and forth between the article, survey questions (supplement) and the dataset resulting from this. The survey questions could also be added to the Pangaea page to make it easier to have everything together.

   A link to the final paper will be added to the Pangaea webpage upon publication. We have added the complete overview of the survey questions that was previously included as a supplement to the manuscript to the Pangaea page in .pdf format.

6. *Orientation of the questionnaire table: Based on the way I would personally analyze the data (often by one question or a combination of questions), I could imagine that a table orientation where participants are listed as rows and questions are listed as columns could be easier for future processing. I would therefore suggest rotating the table.*

   We have changed the orientation of the table.